# Kinetochores attached to microtubule-ends are stabilised by Astrin bound PP1 to ensure proper chromosome segregation

Duccio Conti[1,2], Parveen Gul[1†], Asifa Islam[1†], José M Martín-Durán[1], Richard W Pickersgill[1], Viji M Draviam[1,2]*

[1]School of Biological and Chemical Sciences, Queen Mary University of London, London, United Kingdom; [2]Department of Genetics, University of Cambridge, Cambridge, United Kingdom

**Abstract** Microtubules segregate chromosomes by attaching to macromolecular kinetochores. Only microtubule-end attached kinetochores can be pulled apart; how these end-on attachments are selectively recognised and stabilised is not known. Using the kinetochore and microtubule-associated protein, Astrin, as a molecular probe, we show that end-on attachments are rapidly stabilised by spatially-restricted delivery of PP1 near the C-terminus of Ndc80, a core kinetochore-microtubule linker. PP1 is delivered by the evolutionarily conserved tail of Astrin and this promotes Astrin's own enrichment creating a highly-responsive positive feedback, independent of biorientation. Abrogating Astrin:PP1-delivery disrupts attachment stability, which is not rescued by inhibiting Aurora-B, an attachment destabiliser, but is reversed by artificially tethering PP1 near the C-terminus of Ndc80. Constitutive Astrin:PP1-delivery disrupts chromosome congression and segregation, revealing a dynamic mechanism for stabilising attachments. Thus, Astrin-PP1 mediates a dynamic 'lock' that selectively and rapidly stabilises end-on attachments, independent of biorientation, and ensures proper chromosome segregation.

*For correspondence:
v.draviam@qmul.ac.uk

†These authors contributed equally to this work

Competing interests: The authors declare that no competing interests exist.

## Introduction

Chromosome segregation can not be initiated until all chromosomes are properly attached to microtubules from opposing spindle poles - a state called biorientation. To mediate and monitor chromosome-microtubule attachments, a macromolecular structure, the kinetochore (KT) assembles on centromeric DNA (reviewed in *Conti et al., 2017*; *Musacchio and Desai, 2017*). Kinetochores are first captured along microtubule-walls (immature lateral attachment) and then brought to microtubule-ends (mature end-on attachment), by a multi-step end-on conversion process (*Tanaka et al., 2005*; *Shrestha and Draviam, 2013*). Only end-on attachments can impart pulling forces to segregate chromosomes apart and in the absence of end-on attachments, checkpoint proteins are retained at the kinetochore which prevents premature chromosome segregation (reviewed in *Musacchio and Desai, 2017*; *Cheeseman, 2014*). A highly responsive set of kinases and phosphatases together monitor and signal kinetochore-microtubule (KT-MT) attachment status (reviewed in *Saurin and Kops, 2016*; *Vallardi et al., 2017*). Whether a similar highly responsive kinase-phosphatase feedback loop exists to selectively and rapidly stabilise end-on attachments is not known; this is important to establish as rapid stabilisation of end-on attachments is essential for withstanding microtubule-end mediated pulling forces that may otherwise detach kinetochore-microtubule attachments as soon as they form.

No outer-kinetochore bound phosphatase has been implicated in the 'selective' stabilisation of end-on attachments, although current models show that incorrect attachments can be destabilised by Aurora-B kinase. Aurora-B-mediated phosphorylation of Ndc80 (a core KT-MT linker) destabilises microtubule binding (*Miller et al., 2008*; *Guimaraes et al., 2008*). Thus, reversing the phosphorylation directly may promote attachment stability, but the phosphatase PP1 recruited by KNL1/hSPC105 to counteract Aurora-B for checkpoint silencing (*Nijenhuis et al., 2014*; *Meadows et al., 2011*; *Rosenberg et al., 2011*; *Liu et al., 2010*) is not essential for stabilising attachments (*Shrestha et al., 2017*) or segregation accuracy (*Zhang et al., 2014*). Therefore, rapid stabilisation of end-on attachments as soon as they form, before biorientation, is likely to require KNL1 independent phosphatases that are either retained and/or newly recruited following the formation of end-on attachments.

When end-on attachments form, the checkpoint proteins BubR1, Bub1 and MPS1 that influence attachments are all either removed from or reduced at kinetochores (*Shrestha and Draviam, 2013*; *Kuhn and Dumont, 2017*; *O'Connell et al., 2008*; *Hiruma et al., 2015*). Simultaneously, the Astrin-SKAP complex is recruited to kinetochores (*Shrestha and Draviam, 2013*; *Kuhn and Dumont, 2017*). Astrin-SKAP binds to microtubules through multiple contact points (*Friese et al., 2016*; *Kern et al., 2017*; *Dunsch et al., 2011*; *Tamura et al., 2015*; *Huang et al., 2012*; *Wang et al., 2012*; *Maffini et al., 2009*). In the absence of Astrin-SKAP, the end-on conversion process is disrupted; end-on attachments form but are not stably maintained (*Shrestha et al., 2017*). Hence, determining the mechanisms that control the kinetochore recruitment of Astrin-SKAP can shed light on how cells recognise and maintain end-on attachments. However, the precise site of Astrin recruitment at the kinetochore is not known. Dissecting Astrin-SKAP's role and regulation has not been straightforward as the complex dimerises and binds to both microtubules and kinetochores (*Dunsch et al., 2011*; *Kern et al., 2017*; *Friese et al., 2016*; *Schmidt et al., 2010*). Moreover, its depletion results in a complex phenotype of chromosomes aligning on the metaphase plate but failing to maintain the congressed state due to a spindle collapse, leading to an irreversible mitotic arrest (*Thein et al., 2007*; *Dunsch et al., 2011*; *Kern et al., 2017*; *Manning et al., 2010*; *Shrestha et al., 2017*).

We report a molecular mechanism by which kinetochore attachments made to microtubule-ends, but not microtubule-walls, are rapidly and selectively stabilised independent of biorientation. The microtubule-associated Astrin is recruited selectively to end-on attached kinetochores, prior to the biorientation of kinetochores. We show that this selective recruitment of Astrin relies on the C-terminus of Astrin and occurs proximal to the C-terminus of Ndc80. Next we reveal that Astrin's C-terminus delivers PP1 close to the C-terminus of Ndc80, which in turn promotes Astrin's own enrichment, setting up a positive feedback that rapidly strengthens kinetochore-microtubule bridging. This selective and rapid stabilisation of end-on attachments can be disrupted by mutating the PP1-docking motif of Astrin. In the absence of Astrin:PP1 interaction, kinetochores can not withstand microtubule-mediated pulling forces, which activates the spindle checkpoint and delays anaphase onset. These attachment defects can be reversed by exogenously delivering PP1 near the C-terminus, but not the N-terminus of Astrin, revealing a spatially restricted PP1-mediated 'lock' to stabilise attachments. The defects induced by Astrin:PP1-docking mutants can not be simply overcome by inhibiting Aurora-B (an attachment destabiliser), revealing a previously unrecognised mechanism to stabilise KT-MT attachments. Premature and constitutive delivery of Astrin:PP1 disrupts chromosome congression and mitotic progression, highlighting the need for a dynamic interaction between Astrin and PP1. Thus, by delivering a pool of PP1, Astrin promotes its own enrichment, enabling a highly responsive lock that can dynamically and selectively stabilise end-on attachments and in turn, ensure the timely segregation of chromosomes.

## Results

### Before biorientation, kinetochores bound to MT-ends recruit Astrin

We showed that Astrin is selectively enriched at end-on attached kinetochores and is required to stabilise end-on attachments, independent of biorientation (*Shrestha and Draviam, 2013*; *Shrestha et al., 2017*). The enrichment of Astrin at kinetochores prior to biorientation points to a novel attachment stabilisation mechanism, independent of already known intra- or inter- kinetochore

stretching mediated stabilisation of bioriented attachments (reviewed in *Conti et al., 2017*). To explore this hypothesis, we tracked Astrin recruitment and dynamics at the kinetochores of monopolar spindles that can not biorient kinetochores. Deconvolution time-lapse microscopy revealed two distinct kinetochore localisation patterns for Astrin in cells co-expressing YFP-Astrin and CENP-B-DsRed (centromere marker): YFP-Astrin forms either a low-intensity 'sleeve' decorating the outer-kinetochore and associated microtubule-end or a high-intensity 'crescent' decorating the outer-kinetochore alone (*Figure 1A*). Tracking the fate of Astrin -sleeves and -crescents in cells co-expressing YFP-Astrin and CENP-B-DsRed, showed a gradual conversion of YFP-Astrin sleeve into brighter YFP-Astrin crescent (*Figure 1A* and *Figure 1—figure supplement 1*). To confirm that both Astrin-crescents and sleeves are found at the outer-kinetochore, we analysed YFP-Astrin localisation in cells co-expressing Nuf2-CFP, a bonafide outer-kinetochore marker. Overlap with Nuf2-CFP signal was partial in Astrin-sleeves and complete in Astrin-crescents, confirming the outer-kinetochore localisation of Astrin prior to biorientation (*Figure 1B*).

To pinpoint precisely where the Astrin-SKAP complex is recruited at the kinetochore, we took clues from previous biochemical studies (*Wang et al., 2012*; *Kern et al., 2017*) and tested whether Astrin-SKAP and Ndc80-Nuf2, two complexes bridging the outer-kinetochore and microtubules, are in close proximity using Förster Resonance Energy Transfer (FRET) microscopy. Previously, fragments of the two complexes were shown to bind microtubules in vitro (*Kern et al., 2017*; *Friese et al., 2016*), but their relative orientations were not known. So, we analysed FRET intensities at the kinetochores of cells co-expressing either Astrin-CFP and Ndc80$^{HEC1}$-YFP or YFP-Astrin and Nuf2-CFP, an Ndc80 partner (*Wigge and Kilmartin, 2001*; *Meraldi et al., 2004*; *Ciferri et al., 2005*; *DeLuca et al., 2006*) (*Figure 1C,D* and *Figure 1—figure supplement 2A,B*). FRET signals were prominent only in cells co-expressing C-terminal fluorescent protein tags for both Ndc80 and Astrin (*Figure 1C,D*), indicating that the C-termini of both proteins are within a 5 nm distance, expected for FRET (*Müller et al., 2013*). In contrast, cells co-expressing YFP-Astrin and Nuf2-CFP did not show prominent FRET signals (*Figure 1—figure supplement 2A,B*), confirming that Astrin is recruited near Ndc80 in a specific orientation at the kinetochore.

In summary, the timing and position of Astrin recruitment to the kinetochore is appropriate for directly stabilising end-on attachments independent of biorientation. First, kinetochore-associated Astrin-sleeve can enrich into an Astrin-crescent prior to biorientation. Second, the C-terminus of Astrin resides proximal to the C-terminus of Ndc80 ideally positioned to strengthen kinetochore-microtubule bridging after the formation of end-on attachments (*Figure 1E*).

## Selective recognition of end-on kinetochores requires Astrin's 70a.a tail

To determine how end-on kinetochores are selectively recognised, we searched for the minimal region of Astrin needed for its kinetochore enrichment as crescents. For Astrin recruitment at kinetochores, its C-terminal 694–1193 a.a is sufficient (*Kern et al., 2017*) and 851–1193 a.a is essential (*Dunsch et al., 2011*). At the very C-terminus of Astrin, using structure prediction tools (*Wolf et al., 1997*), we identify a new unstructured region referred here as Astrin tail (*Figure 2A*). Removing 70 amino acids (1123–1193 a.a) of the unstructured region (YFP-Astrin Δ70) abrogated Astrin's kinetochore enrichment as crescents; the deletion mutant was either present as sleeves or completely absent at the kinetochore, but was normally present on spindle microtubules and poles (*Figure 2B, C*; *Figure 2—figure supplement 1A*), revealing a specific role for Astrin tail in recognising end-on kinetochores. Moreover, in cells expressing YFP-Astrin Δ70 mutant (*Figure 2B*), SKAP failed to localise at the kinetochore but not spindle microtubules, revealing an important role for the 70 a.a long Astrin tail in targeting the entire Astrin-SKAP complex to the kinetochore-microtubule interface.

To confirm that the 70 a.a tail of Astrin is specifically required for Astrin function at kinetochores, but not spindle microtubules, we tested whether the Astrin-Δ70 mutant is required for bipolar spindle maintenance. We quantified the proportion of bipolar and multipolar spindles in cells depleted of endogenous Astrin and expressing either Astrin-WT or -Δ70 mutant (*Figure 2D,E*). While Astrin depleted cells displayed multipolar spindles as reported (*Thein et al., 2007*), cells expressing either Astrin-WT or -Δ70 displayed bipolar spindles, indicating a successful rescue of the multipolar spindle phenotype seen in the absence of Astrin. We conclude the kinetochore pool of Astrin-SKAP is dispensable for bipolar spindle maintenance. Thus, the Δ70-mutant uncouples Astrin's role at the kinetochore from the rest of the spindle and serves as a molecular probe to test how Astrin selectively recognises and stabilises end-on attachments.

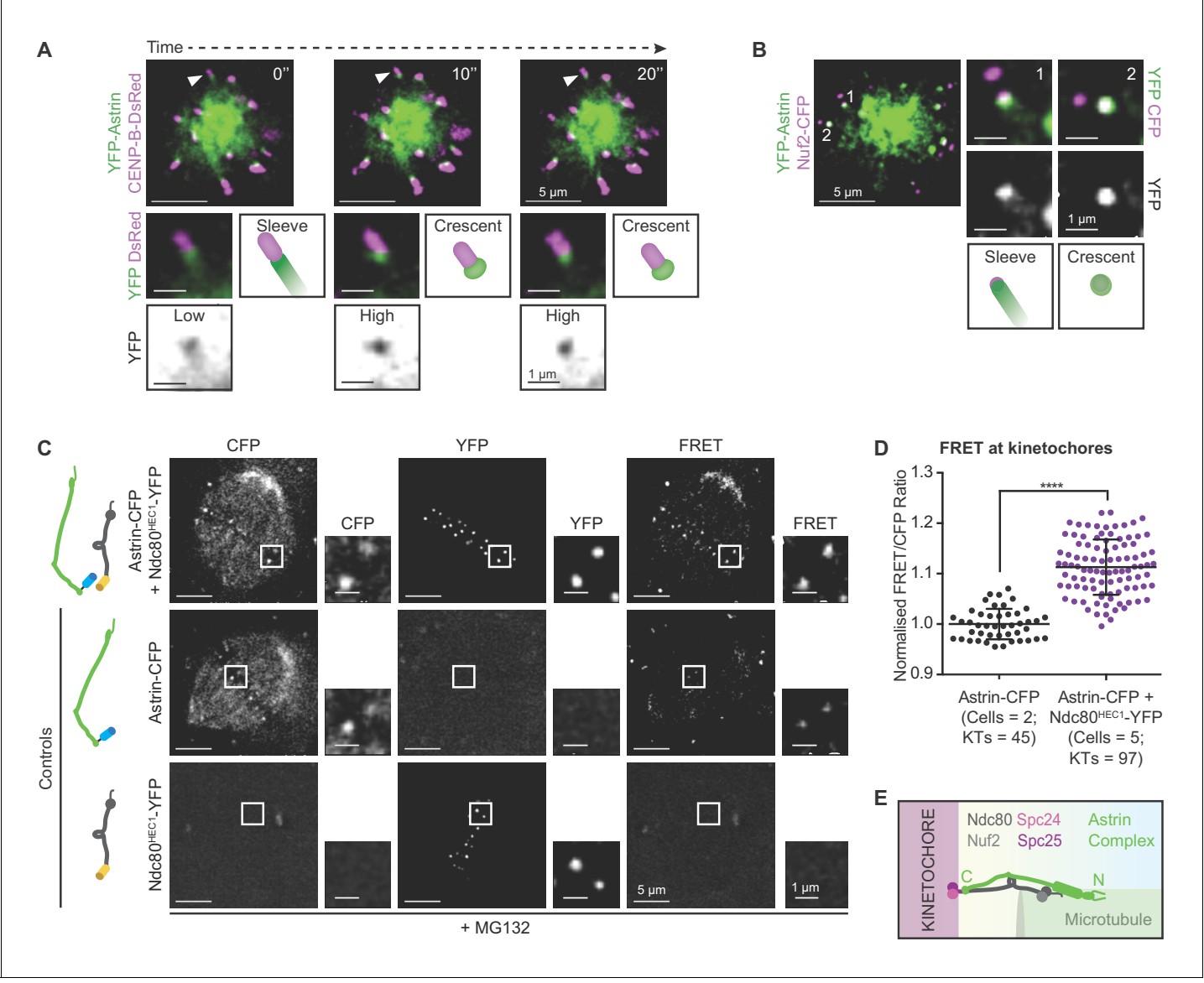

**Figure 1.** Before biorientation, Astrin is recruited to kinetochores proximal to Ndc80. (**A**) Time-lapse images of monopolar spindles in STLC treated HeLa FRT/TO cells co-expressing YFP-Astrin and CENP-B-DsRed show transition of Astrin from sleeve-like to crescent-like kinetochore structure. Each frame is an average-intensity projection of 3 z-planes, 300 nm apart. White arrowhead in main image marks kinetochore magnified in cropped-insets. 'Sleeve' refers to low Astrin signal intensities at kinetochore that extend into the microtubule-end and 'Crescent' refers to relatively high Astrin signal intensities retained exclusively at kinetochore. YFP inset shows inverted image intensities to highlight Astrin-low to Astrin-high transition. (**B**) Representative live-cell image (average-intensity projections of 4 z-planes 200 nm apart) of monopolar spindles show Astrin -sleeves and -crescents in STLC treated HeLa FRT/TO cells coexpressing YFP-Astrin and Nuf2-CFP. Numbers mark kinetochores magnified in cropped-insets. (**C**) Representative live-images of bipolar spindles show FRET emission (excitation$^{CFP}$/emission$^{YFP}$) in MG132 treated HeLa cells expressing either Astrin-CFP or Ndc80$^{HEC1}$-YFP singly or both Astrin-CFP and Ndc80$^{HEC1}$-YFP as indicated (cartoons on left). (**D**) Graph shows FRET/CFP intensity ratios of kinetochores imaged as in (**C**). Each dot represents intensity ratios from one kinetochore. Black bars and whiskers mark average value and standard deviation, respectively, across two experimental repeats. '*' indicates statistically significant difference. Scale as indicated. (**E**) Cartoon shows the orientation of kinetochore-microtubule bridges: the C-term of Astrin resides proximal to the C-term of Ndc80.

The online version of this article includes the following figure supplement(s) for figure 1:

**Figure supplement 1.** Transition of Astrin sleeve into a crescent shape at the kinetochore is associated with changes in Astrin intensities.
**Figure supplement 2.** A stable pool of Astrin localises at metaphase kinetochores.

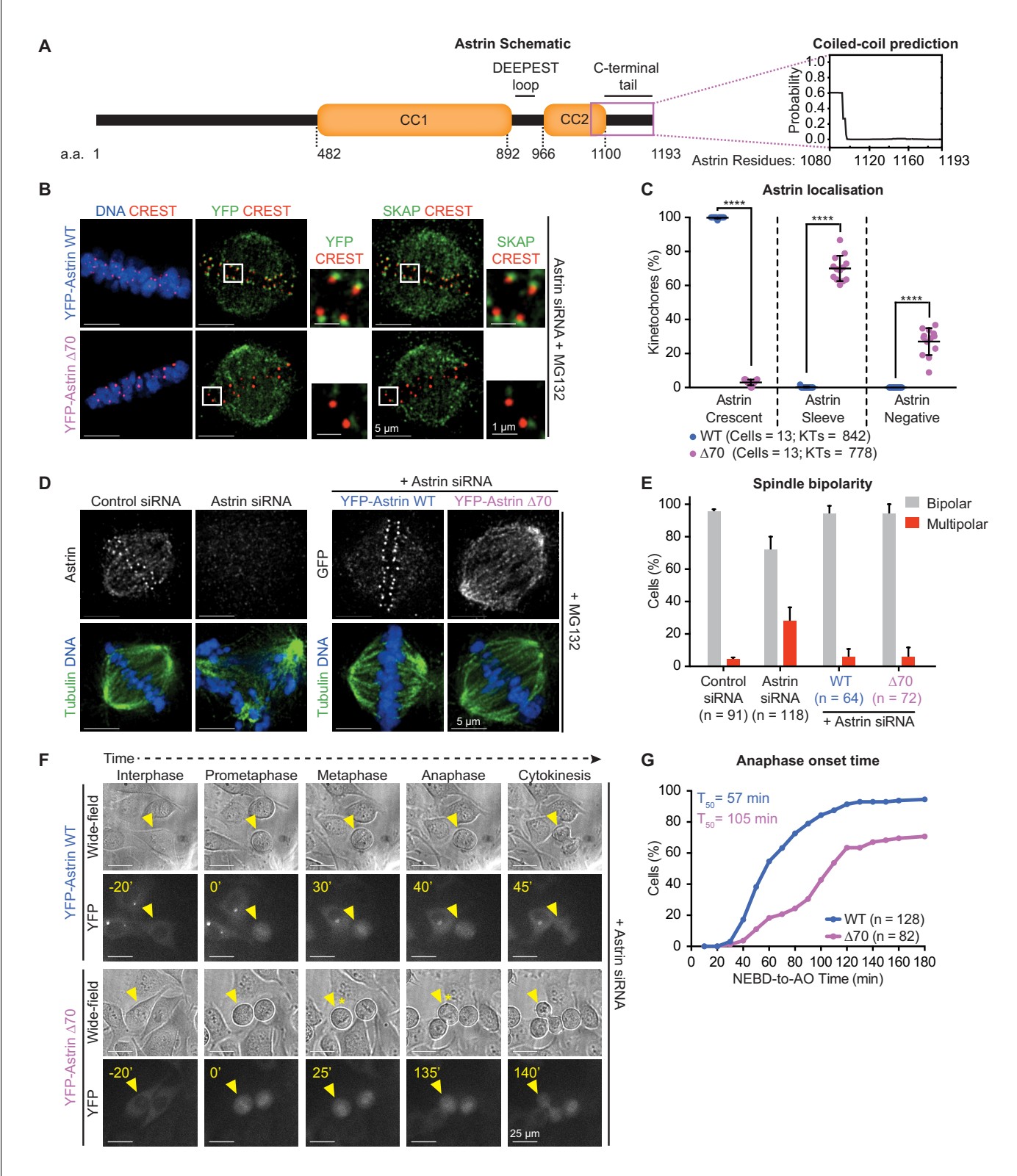

**Figure 2.** Recognition of end-on kinetochores requires Astrin's 70 a.a c-terminal tail. (**A**) Schematic of Astrin protein domains. Graph of predicted coiled-coil or unstructured region in Astrin C-terminus modelled as a dimer using *Multicoil* (***Wolf et al., 1997***) (1080–1193 a.a). (**B**) Representative deconvolved images show YFP-Astrin (WT or Δ70) kinetochore intensities. HeLa cells depleted of endogenous Astrin, expressing siRNA-resistant YFP-Astrin (WT or Δ70) were arrested with MG132, immunostained with antibodies against GFP and SKAP and CREST antisera and stained with DAPI for

*Figure 2 continued on next page*

*Figure 2 continued*

DNA. (C) Graph of percentage of Astrin sleeves or crescents at the outer-kinetochores of YFP-Astrin (WT or Δ70) expressing cells as in (B). Black bars and whiskers mark average value and standard deviation, respectively, across two experimental repeats. '*' indicates statistically significant differences. (D) Representative deconvolved images show the rescue of spindle bipolarity defects in cells depleted of endogenous Astrin expressing an siRNA-resistant YFP-Astrin (WT or Δ70). Astrin depletion was confirmed by comparing levels of endogenous Astrin in Control *versus* Astrin siRNA treated cells. Cells were immunostained with antibodies against either GFP or Astrin (indicated) and Tubulin and co-stained with DAPI for DNA. (E) Bar graph of percentage of bipolar or multipolar spindles in mitotic cells treated as in (D). Bars and whiskers mark average value and standard deviation, respectively, across at least three experimental repeats. (F) Time-lapse images of HeLa FRT/TO cells treated with Astrin siRNA and expressing either YFP-Astrin (WT or Δ70). Yellow triangle indicates the cell tracked; Yellow asterisks highlight prolonged delay between metaphase and anaphase. Cytoplasmic YFP signal was used to assess Nuclear Envelope BreakDown (NEBD), wide-field and YFP images were used to assess bipolar metaphase spindles undergoing anaphase (AO). (G) Cumulative graph of percentage of HeLa FRT/TO cells (as in F) that initiated NEBD and completed AO within time intervals indicated. 'n' refers to cell numbers. $T_{50}$ indicates AO time consumed by at least 50% of mitotic cells. Scale as indicated.

The online version of this article includes the following figure supplement(s) for figure 2:

**Figure supplement 1.** Depletion of endogenous Astrin and conditional expression of Astrin mutants.

Using time-lapse microscopy, we tracked the fate of mitotic cells lacking the 70 a.a tail of Astrin by expressing an siRNA-resistant YFP-Astrin-WT or -Δ70 mutant following siRNA-mediated depletion of endogenous Astrin (*Figure 2—figure supplement 1B,C*). Analysis of the time taken from Nuclear Envelope Break Down (NEBD) to anaphase onset, using YFP-Astrin signal on the spindle, showed a delayed anaphase onset in cells expressing Astrin-Δ70 mutant compared to Astrin-WT; spindle bipolarity and chromosome alignment or congression were however unperturbed in mutant expressing cells (*Figure 2F,G*). In summary, kinetochore-bound Astrin is required for timely anaphase, but not the congression of chromosomes or the maintenance of bipolar spindles. The C-terminal tail of Astrin enables the selective recognition of end-on attached kinetochores and the timely onset of chromosome segregation.

## Withstanding microtubule-end mediated kinetochore pulling requires Astrin's PP1-docking motif

To understand why the 70 a.a tail of Astrin is crucial for timely anaphase onset, we screened for evolutionarily conserved residues in the tail region. This revealed a putative PP1-docking motif RVxF, previously reported in KNL1 (*Bajaj et al., 2018*). Importantly, using the unstructured C-terminal tail, a bioinformatic search could identify Astrin elsewhere in Bilateria (*Figure 3—figure supplement 1* and Source data 1), suggesting a previously unrecognised evolutionarily conserved role for the Astrin-Tail. Modelling Astrin's RVxF bearing peptide sequence 'KLRVMFLEMKN', using a KNL1 peptide conformation bound to PP1 (*Bajaj et al., 2018*) shows the Met in RVMF in an accessible region potentially allowing almost any a.a except for Pro (*Figure 3—figure supplement 2*).

Mutating the PP1-docking motif RVxF into AAAA in Astrin-Tail (referred as, Astrin-4A mutant) disrupts Astrin localisation at end-on kinetochores but not spindle microtubules (see below). The incidence of YFP-Astrin-4A crescents was significantly reduced compared to YFP-Astrin-WT crescents (*Figure 3A,B*; *Figure 3—figure supplement 3A*) in congressed kinetochores of metaphase cells immunostained with anti-GFP antibody and CREST antisera. However, the incidence of Astrin sleeves at congressed kinetochores was significantly increased in cells expressing YFP-Astrin-4A compared to YFP-Astrin-WT (*Figure 3B*), showing a specific failure in Astrin enrichment at kinetochore as crescents. These findings show that the RVxF motif is critical for enriching but not the initial recruitment of Astrin at the kinetochore.

We tested whether reducing the kinetochore pool of Astrin using Astrin-4A mutant could delay anaphase onset. Tracking mitosis in YFP-Astrin-4A expressing cells treated with Astrin siRNA showed a striking delay in anaphase onset, compared to cells expressing YFP-Astrin-WT (*Figure 3C,D* and *Figure 3—figure supplement 3B*). This delay in anaphase onset was despite the normal completion of chromosome congression at the spindle equator (*Figure 3C*). In agreement, MG132-treated metaphase cells depleted of endogenous Astrin and expressing YFP-Astrin-4A maintained a bipolar spindle and aligned chromosomes similar to cells expressing YFP-Astrin-WT (*Figure 3—figure supplement 3C,D*). In conclusion, the PP1 docking RVxF motif is dispensable for spindle bipolarity and chromosome congression but is crucial for the enrichment of Astrin-crescents and normal timing of anaphase.

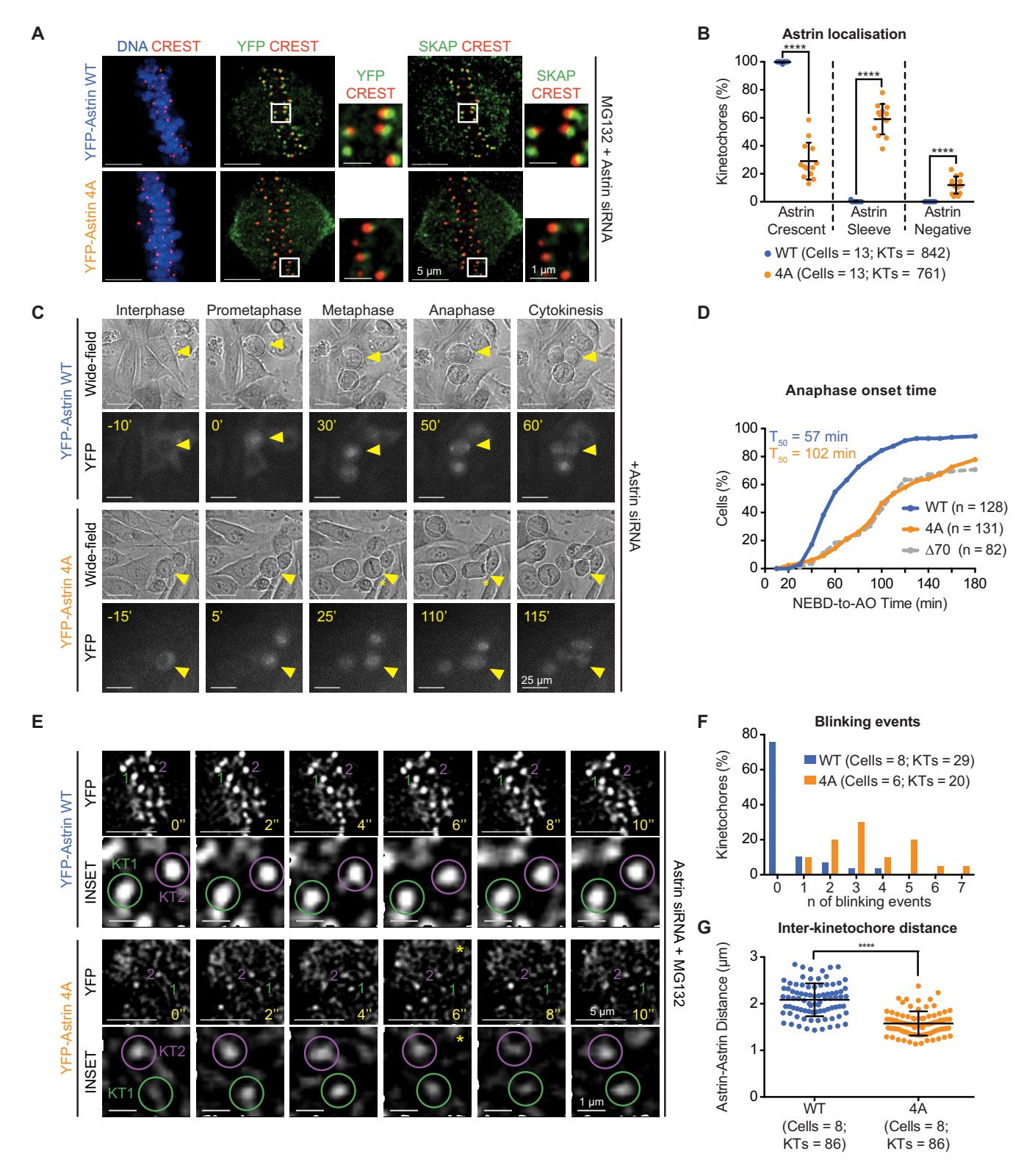

**Figure 3.** Withstanding MT-end mediated KT-pulling needs Astrin's PP1 docking motif. (**A**) Representative images show YFP-Astrin (WT or 4A) kinetochore intensities in HeLa cells depleted of endogenous Astrin and arrested using MG132. Cells were immunostained with antibodies against GFP and SKAP and CREST antisera and stained with DAPI for DNA. (**B**) Graph of percentage of Astrin sleeves or crescents at the outer-kinetochores of YFP-Astrin (WT or 4A), as indicated, in cells treated as in (**A**). Black bars and whiskers mark average value and standard deviation, respectively, from values

*Figure 3 continued on next page*

*Figure 3 continued*

across two experimental repeats. '*' indicates statistically significant differences. (C) Time-lapse images of HeLa FRT/TO cells treated with Astrin siRNA and expressing YFP-Astrin (WT or 4A), as indicated. Yellow triangle indicates the cell tracked; Yellow asterisks highlight prolonged delay between metaphase and anaphase. Cytoplasmic YFP signal was used to assess Nuclear Envelope BreakDown (NEBD), wide-field and YFP images were used to assess bipolar metaphase spindles undergoing anaphase (AO). (D) Cumulative graph of percentage of HeLa FRT/TO cells (as in C) that initiated NEBD (Nuclear Envelope Break-Down) and completed AO (Anaphase Onset) within time intervals indicated. n and $T_{50}$ refer to cell numbers and AO time in 50% of cells, respectively. Astrin Δ70 and WT data are from *Figure 2G*. (E) Time-lapse images of HeLa FRT-TO YFP-Astrin (WT or 4A) siRNA-resistant cells, depleted of endogenous Astrin and arrested with MG132 (1 hr) before imaging. Green and magenta circles highlight Astrin-crescents. KT2 (magenta circle) in YFP-Astrin-4A expressing cell marks an Astrin-crescent gradually dimming into an Astrin-sleeve. Yellow asterisk refers to this blinking event. (F) Frequency graph of distribution of the number of blinking events (as in E) in kinetochores tracked for 60 s. KTs and n refers to the kinetochore numbers and blinking events, respectively. (G) Graph of average distance between Astrin-crescents of kinetochore pairs measured using time-lapse movies as in (E). Black bars and whiskers mark average value and standard deviation, respectively, across at least two experimental repeats. '*' indicates statistically significant differences. Scale as indicated.

The online version of this article includes the following figure supplement(s) for figure 3:

**Figure supplement 1.** Evolutionary conservation of Astrin tail across Bilateria.
**Figure supplement 2.** Model of Astrin peptide based on KNL1 peptide conformation bound to PP1 (PDB code 6CZO).
**Figure supplement 3.** AAAA mutation in Astrin-Tail disrupts Astrin levels at kinetochores but not spindle bipolarity.
**Figure supplement 4.** Inter-centromeric distances reduce following Astrin-4A expression.

To determine the precise reason for delayed anaphase, we performed high-resolution time-lapse imaging of congressed kinetochores in MG132-treated cells depleted of endogenous Astrin and expressing either YFP-Astrin-WT or YFP-Astrin-4A. Analysing the intensity and size of YFP-Astrin signals associated with kinetochore pairs showed dimmer and smaller crescents of Astrin-4A, compared to Astrin-WT (*Figure 3E*). In Astrin-4A expressing cells, dim kinetochore-crescents gradually declined into Astrin-sleeves (referred as, blinking; *Figure 3E*), explaining reduced Astrin-4A crescent intensities in fixed-cells (*Figure 3A* and *Figure 3—figure supplement 3A*). In contrast, Astrin-WT crescents were retained brightly at congressed kinetochores (*Figure 3E*), mediating stable kinetochore-microtubule bridging. This confirms that the PP1 docking motif is required for stably retaining Astrin at kinetochores. Next, we assessed the life-time of Astrin-crescents by counting the number of 'blinking' events (prominent intensity changes) within a 60 s window. Unlike Astrin-WT-expressing cells, Astrin-4A-expressing cells displayed multiple blinking events within a short 60 s window, demonstrating compromised kinetochore life-time for YFP-Astrin-4A (*Figure 3F*). We conclude that the putative PP1-docking motif is not essential for the initial arrival of Astrin sleeves at kinetochores; it is essential for the maintenance of Astrin-crescents, revealing a role for the PP1-docking motif in the stable maintenance of kinetochore-microtubule bridges.

If a reduction in Astrin complex weakens kinetochore-microtubule bridging, we are likely to observe reduced pulling of sister kinetochores by microtubule-end mediated forces. To assess whether sister kinetochores experience normal microtubule-end mediated pulling forces in cells lacking the PP1-docking motif, we measured inter-kinetochore distances in live-cells expressing Astrin-4A mutant. Despite normal kinetochore alignment at the spindle equator, inter-kinetochore distances between Astrin crescents were significantly reduced in cells expressing Astrin-4A compared to Astrin-WT (*Figure 3G*). To confirm the reduction in inter-kinetochore distances following Astrin-4A expression, we performed live-cell imaging of metaphase cells coexpressing CENP-B-DsRed, a centromere marker and YFP-Astrin following Astrin siRNA treatment. Time-lapse movies indicated transient but steep reduction in distances between CENP-B-DsRed pairs of cells expressing YFP-Astrin-4A but not YFP-Astrin-WT (*Figure 3—figure supplement 4A*). Measuring distances between CENP-B-DsRed pairs confirmed a significant reduction in average inter-kinetochore distances of cells expressing YFP-Astrin-4A compared to YFP-Astrin-WT (*Figure 3—figure supplement 4B*), indicative of a failure in microtubule-mediated kinetochore pulling in Astrin-4A-expressing cells.

To determine whether reduced kinetochore life-time of Astrin-4A leads to reduced kinetochore pulling or conversely, an inability to withstand microtubule-mediated pulling leads to a reduction in Astrin-4A at kinetochores, we tested whether a brief treatment with 100 nm Taxol, which pauses microtubule dynamics and reduces microtubule-mediated pulling (*Draviam et al., 2006*; *Shrestha et al., 2014*), would allow the enrichment of Astrin-4A or -Δ70 at kinetochores. Taxol treatment induced no changes in the levels of YFP-Astrin-4A or -Δ70 at kinetochores (*Figure 3—figure*

*supplement 3E*). We conclude that reduced microtubule-mediated pulling is a consequence, rather than the cause, of the reduced life-time of Astrin-4A at kinetochores.

In summary, selective stabilisation of end-on attachments is mediated by the PP1-binding motif of Astrin, which promotes Astrin's own enrichment. Astrin enrichment at end-on kinetochores is likely to further strengthen kinetochore-microtubule bridges as it is required to withstand microtubule-mediated kinetochore pulling and the timely onset of chromosome segregation.

## Astrin dynamically interacts with PP1 phosphatase in vivo and in vitro

We investigated the extent to which Astrin is capable of interacting with PP1 both in vivo and in vitro. PP1γ enriches at the outer-kinetochore (*Trinkle-Mulcahy et al., 2003*). We tested whether YFP-PP1γ (*Trinkle-Mulcahy et al., 2001*) and the C-terminus of Astrin (Astrin-CFP) are proximal to each other using FRET in live-cells. Cells co-expressing Astrin-CFP and YFP-PP1γ displayed prominent FRET signals only on those kinetochores that retained both Astrin-CFP and YFP-PP1γ (cyan arrow; *Figure 4A*). We confirmed that these FRET signal intensities at kinetochores were well above the low-level of signal bleedthrough observed in cells expressing Astrin-CFP alone (*Figure 4A,B*). Measuring the ratio of intensities between the FRET and CFP (FRET-donor) channels confirmed increased FRET in kinetochores of cells coexpressing Astrin-CFP and YFP-PP1γ compared to those expressing Astrin-CFP alone (*Figure 4B*). As expected, a reduction in Astrin-CFP intensities (FRET-donor depletion) was also observed in some kinetochores that displayed high FRET signals (magenta arrow; *Figure 4A*). These studies show that the C-terminus of Astrin and PP1γ are recruited at the kinetochore within a FRET distance of 5 nm between the CFP and YFP probes (*Müller et al., 2013*), suggesting a direct interaction between the tail of Astrin and PP1 at the outer kinetochore. Similar analysis of FRET ratios at spindle microtubules showed a slightly reduced FRET signal at spindle microtubules compared to kinetochores (compare *Figure 4B,C*). Finally, no FRET signals were observed on kinetochores with Astrin-CFP but not YFP-PP1γ (yellow arrow; *Figure 4A*), revealing the dynamic nature of Astrin:PP1 interaction. We conclude that Astrin and PP1 interact at the outer-kinetochore, and this interaction is dynamic.

To investigate whether Astrin and PP1 interact in vitro, we recombinantly produced and immobilised GST-PP1γ on Glutathione beads and incubated the beads with mitotically enriched lysates of HeLa cells treated with either Astrin or Control siRNA (*Figure 4—figure supplement 1A*). In these pulldown assays, we used recombinantly produced GST immobilised on Glutathione beads as a negative control (*Figure 4D*). Immunoblotting using anti-Astrin antibodies showed that Astrin isoforms can interact with GST-PP1γ, but not GST, in Control siRNA treated mitotic cell lysates, revealing a specific interaction between Astrin and PP1γ (*Figure 4E*). The extent of Astrin pulldown by GST-PP1 is higher while using lysates of cells treated with Control siRNA compared to Astrin siRNA. Moreover, quantifying the ratio of Astrin band intensities in pull down lane relative to input lane of control siRNA treated cell lysates confirm that Astrin interacts with GST-PP1γ, but not GST, in two independent repeats of recombinant protein purification and pull down experiments (*Figure 4F* and *Figure 4—figure supplement 1B*). To confirm the specificity of the anti-Astrin antibody in immunoblots, we compared lysates of unsynchronised or mitotically enriched cells treated with Control or Astrin siRNA (*Figure 4—figure supplement 1C*) and confirmed all three bands to be specific to Astrin in mitotic cell lysates (see green arrowheads in *Figure 4—figure supplement 1C*).

In summary, in vivo FRET studies reveal a dynamic and spatially restricted interaction between Astrin and PP1 at kinetochores. In vitro pull-down studies show a specific interaction between GST-PP1 but not GST. Based on these in vivo and in vitro studies we conclude that Astrin and PP1 interact with each other.

## RVxF motif in Astrin contributes to Astrin-PP1 interaction

To test whether the evolutionarily conserved RVMF motif in Astrin is responsible for Astrin-PP1 interaction, we compared FRET extent between YFP-PP1γ and Astrin-CFP WT or Astrin-CFP 4A mutant proteins (*Figure 4—figure supplement 2A*). We could not assess FRET at all kinetochores of cells expressing the Astrin-4A mutant as the mutant does not present a sustained kinetochore localisation (see also, *Figure 3*). However, using those timepoints when the Astrin-4A mutant was transiently enriched at the kinetochore, we could quantitatively establish that the Astrin-CFP 4A mutant expressing cells present severely reduced FRET signals compared to Astrin-WT expressing cells

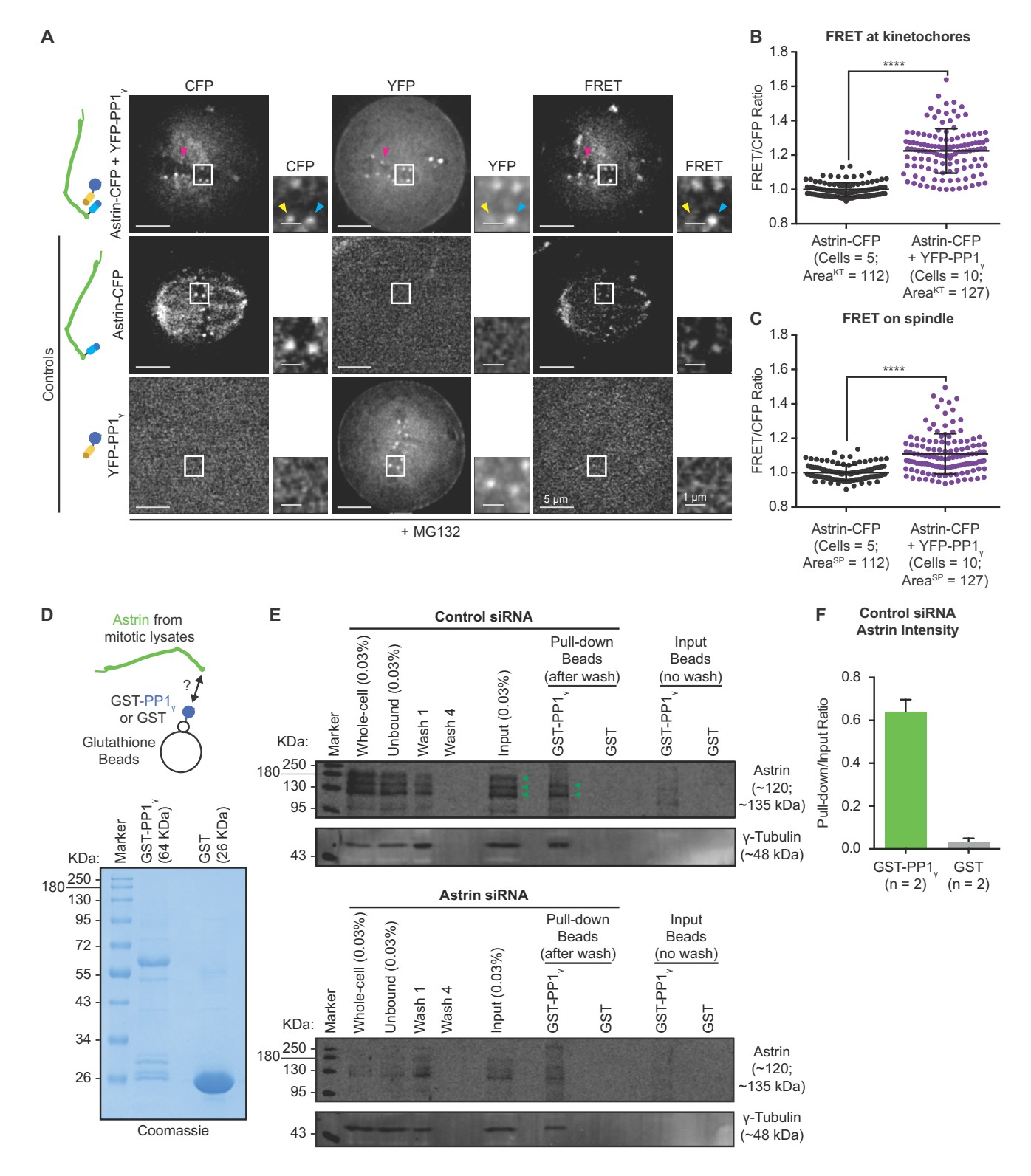

**Figure 4.** Kinetochore bound Astrin dynamically interacts with PP1 phosphatase. (**A**) Representative live-cell images show FRET emission (excitation^CFP/ emission^YFP) signals of HeLa cells expressing either YFP-PP1γ or Astrin-CFP singly or both YFP-PP1γ and Astrin-CFP and treated with MG132 for 1 hr before imaging. Cartoons of fusion proteins expressed in each condition are shown on the left. Yellow arrowhead highlights a CFP-positive and YFP-negative kinetochore not presenting FRET emission signal. Cyan arrowhead marks FRET-positive kinetochore. Magenta arrowhead highlights CFP-low

*Figure 4 continued on next page*

*Figure 4 continued*

and YFP-positive kinetochore presenting FRET emission signal and CFP donor depletion. (B–C) Graphs show the normalised FRET/CFP intensity ratio at kinetochores (B) and spindle microtubules (C), respectively, measured from time-lapse movies as shown in (A). Each dot represents a value from a kinetochore (KT) or a spindle (SP) area of size 0.066 μm². Black bars and whiskers mark average value and standard deviation, respectively, from kinetochores or spindle regions across two experimental repeats. '*' indicates statistically significant differences. Scale as indicated. (D) Coomassie stained gel shows recombinant GST-PP1γ or GST alone, immobilised on Glutathione beads, used for pull-down assays. (E) Representative immunoblot of pull-down assays shows the interaction of GST-PP1γ, but not GST, with Astrin in lysates of mitotically synchronised HeLa cells. Cells were treated with Astrin or Control siRNA, as indicated, and exposed to STLC for 24 hr prior to lysate generation. Immunoblot was probed with antibody against Astrin (top) or gamma-Tubulin (bottom, positive control). Immunoblot is representative of two independent pulldown studies. Astrin-PP1 interaction is more prominent in Control siRNA treated cells compared to Astrin siRNA treated cells. Gamma-tubulin (positive control for PP1 interaction) can be found in both Control or Astrin siRNA treated cells. Immunoblot shows whole cell lysate (WCL) and fractions of unbound and wash supernatents from samples exposed to GST-PP1γ (green arrowheads mark Astrin). For methodology details see *Figure 4—figure supplement 1A*. (F) Graph of ratio of Astrin intensities in the GST-PP1 or GST pull-down lane relative to input lysate lane. Areas used for Astrin intensity measurements are shown in *Figure 4— figure supplement 1B*. n = 2 refers to number of independent repeats of recombinant protein purification and pulldown studies.

The online version of this article includes the following figure supplement(s) for figure 4:

**Figure supplement 1.** Astrin purifies with GST-PP1γ.
**Figure supplement 2.** FRET signals at kinetochores reduced in cells coexpressing YFP-PP1γ and Astrin-CFP-4A compared to Astrin-CFP-WT.

(*Figure 4—figure supplement 2B*). These live-cell studies show that the Astrin-CFP 4A mutant does not interact normally with YFP-PP1γ at kinetochores.

## Stable maintenance of end-on attachments prior to biorientation requires Astrin's PP1-docking motif

Reduced microtubule-mediated pulling of kinetochores in Astrin-4A expressing cells (*Figure 3G* and *Figure 3—figure supplement 4A,B*) suggests an inability to maintain stable end-on attachments. To directly assess whether the RVxF motif in Astrin is important for the maintenance of end-on attachments, we used a quantitative assay that has previously allowed us to clearly identify the stability of three different attachment steps of the end-on conversion process: kinetochores unattached to microtubules; kinetochores attached laterally to microtubule-walls; and kinetochores attached end-on to microtubule-ends (*Shrestha and Draviam, 2013*; *Shrestha et al., 2017*). We quantified the proportion of detached, laterally attached or end-on attached kinetochores in monopolar spindles of cells expressing YFP-Astrin-WT, −4A or -Δ70 following the depletion of endogenous Astrin. To analyse kinetochore-microtubule attachment status, we immunostained cells using an antibody against Tubulin and the CREST antisera (centromere marker) and imaged using deconvolution microscopy. Compared to cells expressing Astrin-WT, cells expressing Astrin mutant (4A or Δ70) displayed significantly fewer end-on attached kinetochores. Importantly, laterally-attached kinetochores, and not detached kinetochores, were significantly increased in Astrin mutant-expressing cells (*Figure 5A,B*). Thus, Astrin mutant-expressing cells are able to retain kinetochores along microtubule-walls but they cannot maintain kinetochores at microtubule-ends, similar to Astrin depleted cells (*Shrestha et al., 2017*). This reveals that the kinetochore pool of Astrin is important for the selective stabilisation of end-on attachments. In agreement, the incidence of Astrin-crescents was dramatically reduced in monopolar spindles of cells expressing YFP-Astrin-4A or -Δ70, compared to cells expressing YFP-Astrin-WT (*Figure 5C*). These monopolar spindle studies together show that Astrin:PP1 interaction is required specifically to maintain end-on, but not lateral, attachments independent of biorientation status. Thus, although Astrin-mediated delivery of PP1 is not required for chromosome congression (*Figure 2D,E* and *Figure 3—figure supplement 3C,D*), it is crucial for the stable maintenance of end-on attachments (*Figure 5B*).

To assess whether the inability to maintain end-on attachments triggers the spindle checkpoint, which would explain delayed anaphase in PP1-docking mutant expressing cells (*Figures 2* and *3*), we analysed the status of checkpoint proteins in congressed kinetochores of cells expressing YFP-Astrin-WT, −4A or -Δ70. As expected, immunostaining using antibodies against the checkpoint proteins, Mad2 or ZW10 showed very low levels of Mad2 or Zw10 in YFP-Astrin WT expressing metaphase cells (*Figure 5D,E* and *Figure 5—figure supplement 1A*). In contrast, metaphase cells expressing YFP-Astrin-4A or -Δ70 mutant displayed prominent Mad2 or ZW10 signals in a subset of congressed kinetochores (*Figure 5D,E* and *Figure 5—figure supplement 1A*). These data indicate

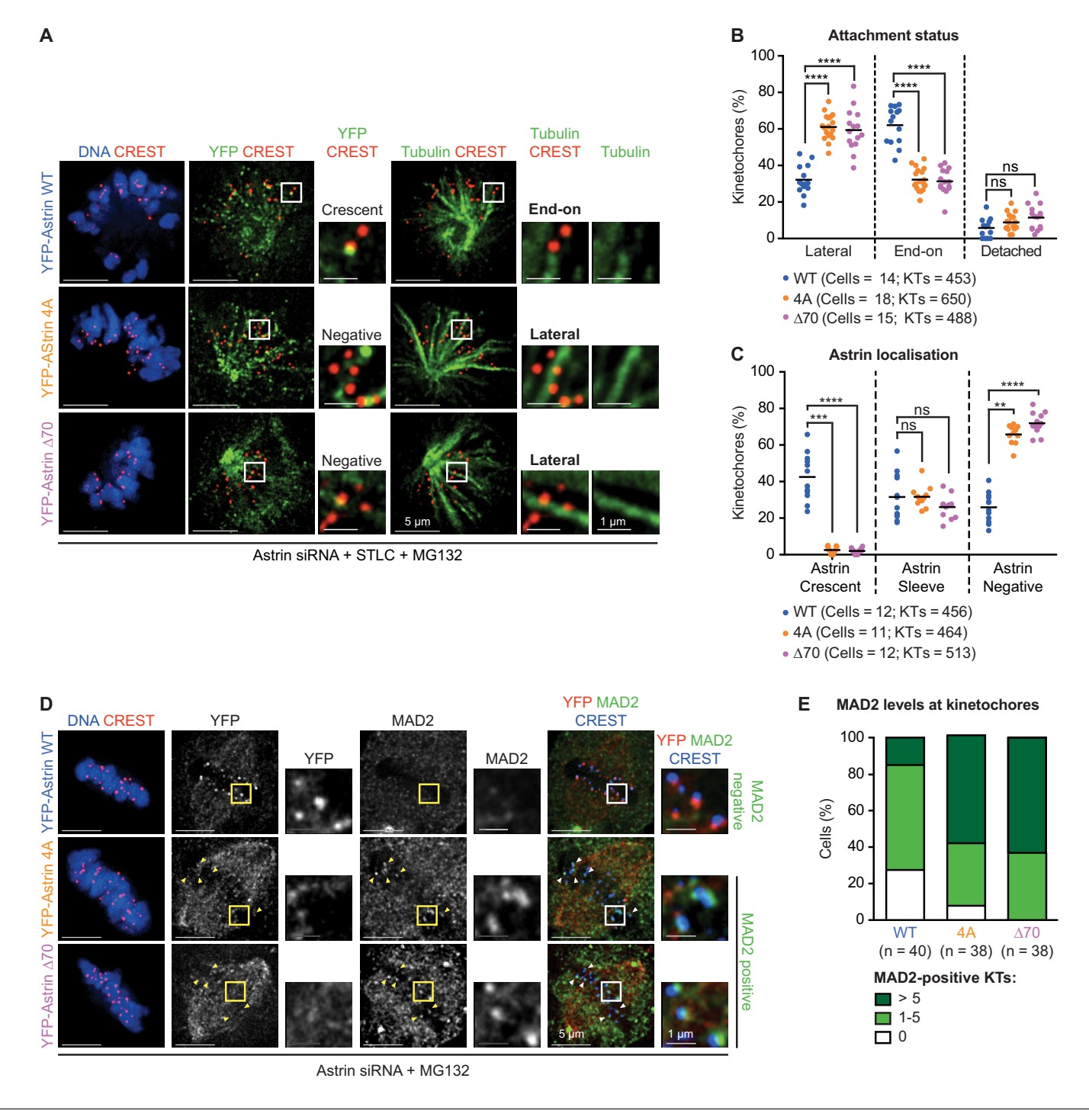

**Figure 5.** Maintenance of end-on attachments before biorientation requires Astrin's PP1 docking motif. (**A**) Representative deconvolved images show Astrin localisation and KT-MT attachment status of HeLa cells, treated with Astrin siRNA, expressing YFP-Astrin (WT, 4A or Δ70). Cells were exposed to either Monastrol or STLC for 2 hr and MG132 for 15' prior to immunostaining with antibodies against GFP or Tubulin and CREST anti-sera and co-staining with DAPI for DNA. Cropped images highlight Astrin localisation and KT-MT attachment status (lateral *versus* end-on). Negative and Crescent refer to the absence and presence of Astrin-crescents, respectively. (**B**) Graph of percentage of lateral, end-on and detached kinetochores in monopolar spindles of YFP-Astrin (WT, 4A or Δ70) expressing cells treated as in (**A**). Each circle represents percentage of kinetochores from one cell. KTs refer to total number of kinetochores assessed. Black bar marks average values from at least three independent experiments. '*' and 'ns' indicate statistically significant and insignificant differences, respectively. (**C**) Graph of percentage of kinetochores presenting sleeve-like or crescent-like Astrin signals in monopolar spindles of cells expressing YFP-Astrin (WT, 4A or Δ70 mutant) treated as in (**A**). Each dot represents a value from one cell. Black bar marks

*Figure 5 continued on next page*

Figure 5 continued
average values from at least three independent experiments. (**D**) Representative deconvolved images show MAD2 levels at kinetochores of HeLa cells depleted of endogenous Astrin and transiently expressing YFP-Astrin (WT, 4A or Δ70). Cells arrested in metaphase using MG132 for 1 hr were immunostained with antibodies against GFP or MAD2 and CREST antiserum and co-stained with DAPI for DNA. Cropped images highlight MAD2 or YFP-Astrin levels at kinetochores identified using CREST antisera. Yellow arrowheads in uncropped images highlight representative kinetochores staining positive for MAD2. (**E**) Bar graph of MAD2-positive kinetochores in metaphase cells expressing YFP-Astrin (WT, 4A or Δ70) treated as in (**D**). Number (n) of cells indicated. Values were obtained from three independent repeats. Scale as indicated.

The online version of this article includes the following figure supplement(s) for figure 5:

**Figure supplement 1.** Checkpoint signalling and chromosome congression efficiency in Astrin:PP1 docking mutant expressing cells.
**Figure supplement 2.** Status of DSN1 Ser100 phosphorylation in Astrin:PP1 docking mutant expressing cells.

an increased incidence of spindle checkpoint signalling in a small proportion of congressed kinetochores in cells expressing either of the two Astrin mutants lacking the PP1-docking motif. Only a few, but not all, kinetochores recruited checkpoint proteins in cells expressing Astrin-4A or -Δ70. As expected from low resolution time-lapse studies (*Figures 2* and *3*), chromosome alignment at the spindle equator (i.e., congression efficiency) is not severely perturbed in Astrin mutant expressing cells (*Figure 5—figure supplement 1B,C*). Thus, the stable maintenance of end-on attachments, but not chromosome congression, requires the evolutionarily conserved PP1-docking motif in Astrin.

## DSN1 phosphorylation is higher on kinetochores of cells expressing Astrin:PP1 docking mutants

To assess the potential downstream phospho-targets of Astrin-PP1 at kinetochores, we tested whether Astrin mutant expression induces a change in the levels of phospho-DSN1, expected to be in the vicinity of Astrin C-terminus. For this study, we used phosphorylation specific antibodies reported previously (*Welburn et al., 2010*). Immunostaining of cells using phospho-specific antibodies against DSN1 pSer100 show that the levels of DSN1 pSer100 are reduced at kinetochores in cells expressing Astrin WT compared to Astrin 4A or -Δ70 mutant (*Figure 5—figure supplement 2A*). Quantitative analysis of phospho-signal positive kinetochores and integrated kinetochore signal intensities confirm an increase in phosphorylated DSN1 signals in cells expressing Astrin 4A or -Δ70 compared to Astrin WT (*Figure 5—figure supplement 2B and C*). These findings show changes in Dsn1 phosphorylation status as a direct or indirect downstream target of Astrin-PP1 at kinetochores.

## KT-MT attachment stability depends on spatially defined delivery of PP1

We hypothesised that if the delivery of a phosphatase is needed to enrich Astrin and stabilise end-on attachments, then exogenous tethering of PP1γ at the C-terminus of Astrin tail mutants should rescue mutant localisation and associated attachment defects. To test this hypothesis, we analysed whether the kinetochore localisation defect of GFP fused Astrin-4A or -Δ70 can be rescued by co-expressing PP1γ fused to a GFP binding protein, GBP (mCherry-GBP-PP1γ; *Rothbauer et al., 2008*). Cells co-expressing Astrin-GFP and mCherry-GBP-PP1γ were arrested in metaphase and immunostained using antibodies against GFP and mCherry to assess Astrin-GFP enrichment at kinetochores. In these rescue studies, the introduction of a C-terminal GFP tag slightly reduced the kinetochore levels of Astrin-GFP compared to N-terminally tagged YFP-Astrin (compare *Figure 6A* to *Figure 6—figure supplement 1A* and *Figure 6B* to *Figure 6—figure supplement 1B*; *Figure 6—figure supplement 1C*). Nevertheless, co-expressing mCherry-GBP-PP1γ fully rescued the kinetochore enrichment defect in both Astrin mutants lacking the PP1-docking motif: both Astrin-Δ70-GFP and Astrin-4A-GFP were enriched as bright crescents at the kinetochore when co-expressed with mCherry-GBP-PP1γ (*Figure 6A,B*). Importantly, no rescue of the localisation defect was observed when Astrin-Δ70 or −4A was fused to an N-terminal YFP tag in mCherry-GBP-PP1γ co-expressing cells (*Figure 6—figure supplement 1A,B*). These studies show that the kinetochore enrichment of Astrin is dependent on spatially-defined delivery of PP1 phosphatase at Astrin's C-terminus (*Figure 6C*), which we show is proximal to the C-terminus of Ndc80 (*Figure 1*).

We next tested whether the exogenous delivery of PP1γ can restore the function of PP1-docking mutants and stabilise attachments. For this purpose, we tested kinetochore-microtubule attachment

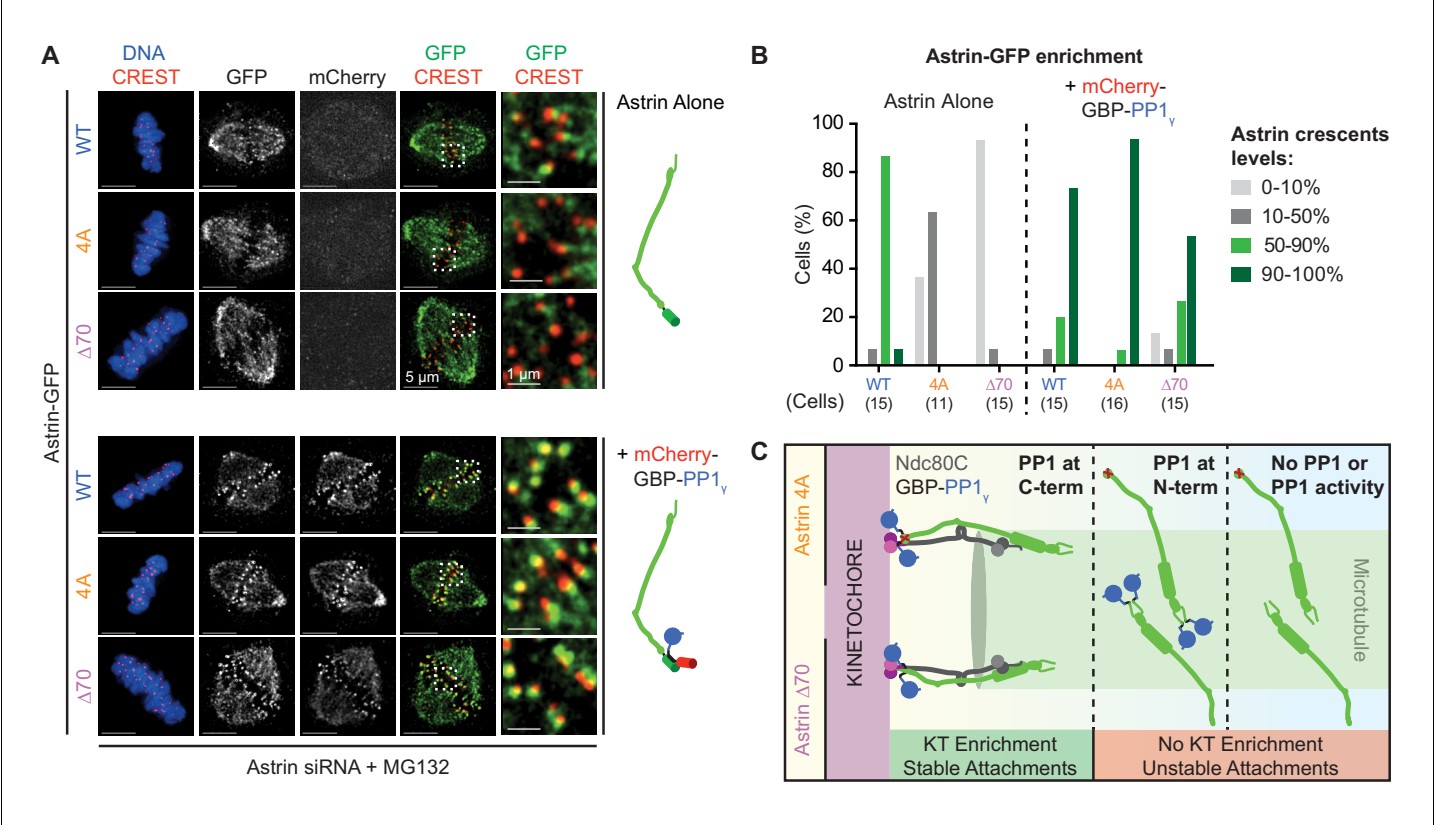

**Figure 6.** Spatially defined delivery of PP1 controls Astrin localisation and function. (**A**) Representative deconvolved images show Astrin-GFP intensities at kinetochores of cells treated with Astrin siRNA and expressing Astrin-GFP (WT, 4A or Δ70) alone or along with mCherry-GBP-PP1$_\gamma$ (cartoons on right). Cells arrested in MG132 were immunostained with antibodies against GFP and mCherry along with CREST antisera and stained with DAPI for DNA. Cropped images are magnified areas boxed in dashed white lines. (**B**) Bar graph of percentage of cells displaying Astrin-crescents on 0–10%, 10–50%, 50–90% or 90–100% of kinetochores in cells expressing Astri-GFP (WT, 4A or Δ70) and mCherry-GBP-PP1$_\gamma$ and treated as in (**A**). Number of cells is indicated. (**C**) Cartoon summarises the consequence of abrogating or spatially restricting the delivery of phosphatase by Astrin. The PP1-docking mutants (4A and Δ70) are enriched at kinetochores only when active PP1 is tethered at Astrin's C-terminus. Astrin:PP1 enrichment at kinetochores is required for the maintenance of cold-stable kinetochore-microtubule attachments. Scale as indicated.

The online version of this article includes the following figure supplement(s) for figure 6:

**Figure supplement 1.** PP1-delivery away from the C-terminus of Ndc80 is insufficient to rescue Astrin mutant localisation.
**Figure supplement 2.** Active PP1$_\gamma$ is required for Astrin localisation and function .
**Figure supplement 3.** SKA3 localisation is not abrogated due to the lack of Astrin.

stability in cells co-expressing mCherry-GBP-PP1γ with either Astrin-4A-GFP or Astrin-Δ70-GFP using a cold stability assay. Upon a brief exposure to cold, cells depleted of Astrin depolymerise all spindle microtubules due to unstable kinetochore-microtubule attachments (*Thein et al., 2007*; *Dunsch et al., 2011*). Similarly, Astrin-depleted cells expressing either Astrin-Δ70 or −4A alone display cold unstable kinetochore-microtubule attachments (*Figure 6—figure supplement 2*, left). In contrast, cells co-expressing mCherry-GBP-PP1γ and Astrin-Δ70-GFP or −4A-GFP display cold stable kinetochore-microtubule attachments, demonstrating a successful restoration of attachment stability in Astrin-Δ70 or −4A mutant expressing cells by delivering PP1γ (*Figure 6—figure supplement 2*, right).

In summary, these findings show that the delivery of PP1 near the C-terminus of Ndc80 is sufficient to promote the kinetochore enrichment of Astrin-SKAP complex and stabilisation of end-on attachments (*Figure 6C*).

The SKA complex can also recruit PP1 to the outer-kinetochore (*Sivakumar et al., 2016*) near the Ndc80 complex (*Janczyk et al., 2017*; *Helgeson et al., 2018*). We tested whether a failure to deliver PP1 by Astrin interferes with the kinetochore levels of the SKA complex, using SKA3 signal as

a readout. Depletion of Astrin did not abrogate SKA3 localisation at congressed kinetochores (*Figure 6—figure supplement 3A*). Similarly, in Astrin -Δ70 or −4A expressing cells, depleted of endogenous Astrin, SKA3 recruitment was unperturbed at congressed kinetochores (*Figure 6—figure supplement 3B*). In conclusion, SKA complex enrichment is not linked to Astrin enrichment at the kinetochore. Thus, the failure to deliver PP1 by Astrin weakens KT-MT attachment stability without abrogating SKA complex recruitment at kinetochores.

## Both Astrin:PP1 and Aurora-B regulate attachment stability

Aurora-B kinase controls microtubule stability by phosphorylating several kinetochore proteins (*Hauf et al., 2003*; *Ditchfield et al., 2003*; *Shrestha et al., 2017*; *Miller et al., 2008*; *Tanaka et al., 2002*; *Welburn et al., 2010*). Reversing Aurora-B phosphorylation is thought to be sufficient for stabilising kinetochore-microtubule attachments (reviewed in *Kelly and Funabiki, 2009*; *Lampson and Cheeseman, 2011*). We tested whether the delivery of PP1 by Astrin is required to (i) simply reverse Aurora-B-kinase mediated phospho-events or (ii) directly stabilise end-on attachments. To distinguish between the two scenarios, we first tested whether inhibiting Aurora-B can overcome the need for Astrin-PP1 interaction in stabilising end-on attachments (*Figure 5A,B*). Inhibition of Aurora-B in monopolar spindles will block the destabilisation of attachments leading to stable end-on attachments (*Lampson et al., 2004*), allowing Astrin enrichment at kinetochores (*Schmidt et al., 2010*; *Shrestha et al., 2017*). However, despite the inhibition of Aurora-B, there was no kinetochore enrichment of 4A or Δ70 mutants in monopolar spindles (*Figure 7A,B* and *Figure 7—figure supplement 1A*). In agreement, Aurora-B inhibited cells expressing Astrin-4A or -Δ70 displayed a higher proportion of laterally-attached kinetochores compared to Astrin-WT expressing cells (*Figure 7C*). These findings show that inhibiting Aurora-B does not stabilise end-on attachments in the absence of Astrin:PP1 interaction, showing the significance of Astrin:PP1 in stabilising KT-MT attachments.

We next tested whether Astrin-mediated delivery of PP1 is sufficient to stabilise end-on attachments. For this purpose, we took advantage of the C-terminally tagged Astrin-Δ70 and −4A mutants and co-expressed mCherry-GBP-PP1γ. For a quantitative analysis of KT-MT attachment status, we used STLC-treated monopolar spindles and blocked premature mitotic exit using MG132 in the presence of the Aurora-B inhibitor, ZM447439. Co-expression of mCherry-GBP-PP1γ rescued the kinetochore localisation defect of both mutants (Astrin-4A-GFP and Astrin-Δ70-GFP) in monopolar spindles of cells treated with Aurora-B inhibitor (compare *Figure 7D* and *Figure 7—figure supplement 1A* and *Figure 7E*). In addition, the co-expression of mCherry-GBP-PP1γ increased the incidence of end-on attachments in Aurora-B inhibited cells expressing either Astrin-4A-GFP or Astrin-Δ70-GFP (*Figure 7D*; compare *Figure 7F and C*). These data show that the stabilisation of end-on attachments following the inhibition of Aurora-B is acutely dependant on the delivery of PP1 by Astrin. In the presence of Aurora-B activity (DMSO-treated cells), the co-expression of mCherry-GBP-PP1γ is insufficient to rescue both the kinetochore localisation and attachment defects of Astrin-4A-GFP or Astrin-Δ70-GFP expressing cells (*Figure 7E,F* and *Figure 7—figure supplement 1B,C*), showing that both Astrin-PP1 and Aurora-B control KT-MT attachment status.

## Astrin:PP1 acts as a dynamic 'lock' that stabilises attachment and ensures normal anaphase

We wondered whether a dynamic delivery of PP1 by Astrin is essential to prevent premature stabilisation of incorrect attachments. To test this hypothesis, we studied the consequence of premature and constitutive delivery of PP1 by Astrin during the conversion of a monopolar spindle into a bipolar one. We expressed Astrin-WT-GFP alone or together with mCherry-GBP-PP1γ in Astrin siRNA treated cells, and exposed these cells to STLC for 4 hr to form monopolar spindles following which we prematurely stabilised end-on attachments by adding Aurora-B inhibitor and MG132 (*Figure 8A - Figure 8—figure supplement 1A*). When STLC and Aurora-B inhibiting drugs were washed off, cells depleted of endogenous Astrin and expressing Astrin-WT-GFP, disassembled monopolar asters normally to build bipolar spindles within 2 hr (*Figure 8A,B*). In these cells, 45 min after STLC-release, confirming normal disassembly of spindle and KT-MT attachments, the kinetochore enrichment of Astrin-WT-GFP was not obvious (*Figure 8—figure supplement 1E*). In contrast, 45 min after STLC-release, cells coexpressing Astrin-WT-GFP and mCherry-GBP-PP1γ displayed Astrin-WT-GFP enrichment on paired kinetochores (*Figure 8—figure supplement 1E*), confirming a sustained kinetochore

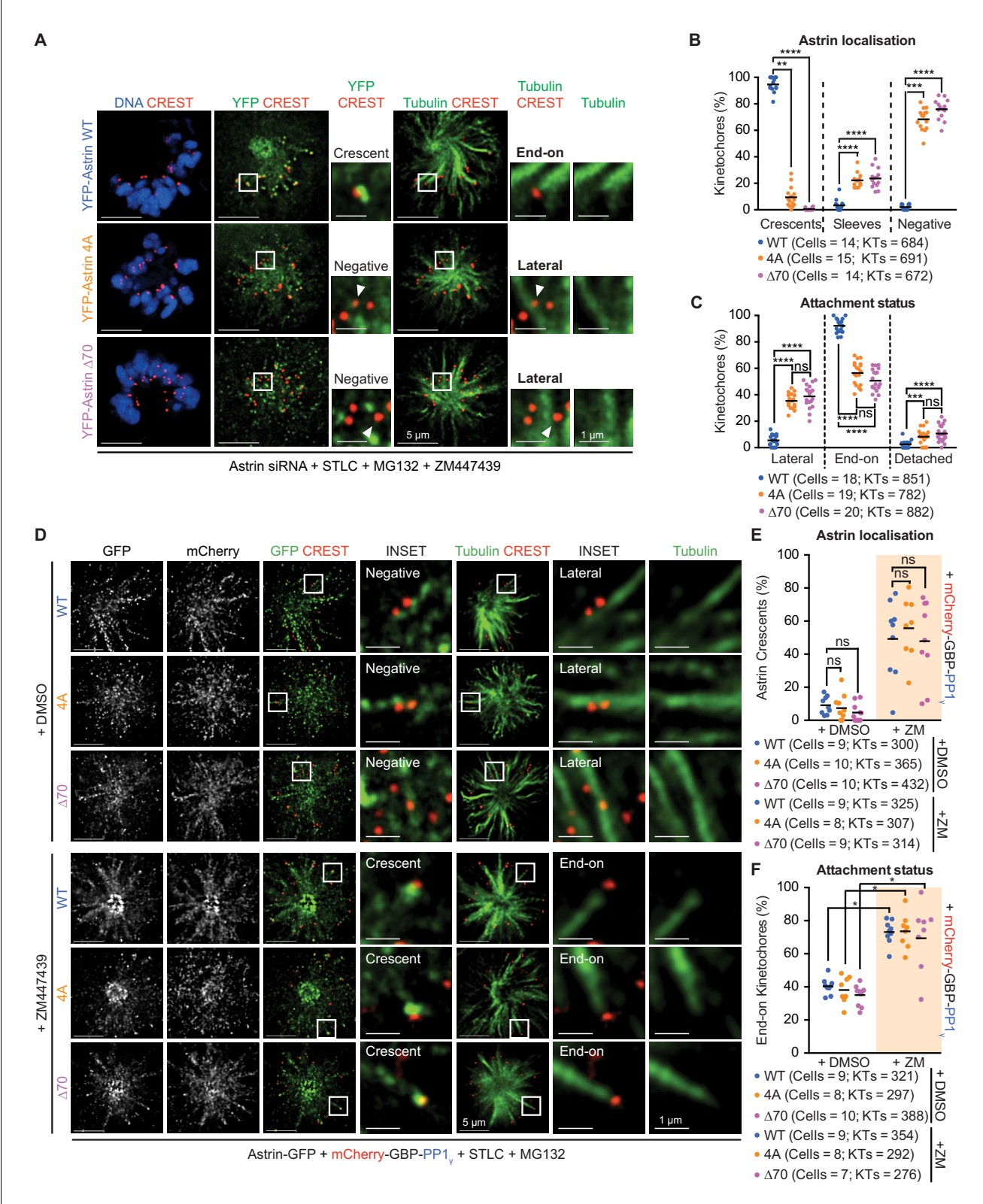

**Figure 7.** Astrin-PP1 and Aurora-B control attachment stability through distinct steps. (**A**) Representative deconvolved images show Astrin localisation and KT-MT attachment status in monopolar spindles of cells depleted of endogenous Astrin expressing YFP-Astrin (WT, 4A or Δ70). STLC treated cells were exposed to MG132 and ZM447439 prior to immunostaining with antibodies against GFP and Tubulin and CREST antisera and staining with DAPI for DNA. Cropped images are areas boxed in white. (**B and C**) Graphs of percentage of kinetochores with Astrin-sleeves or -crescents (**B**) or lateral,

*Figure 7 continued on next page*

*Figure 7 continued*

end-on or detached kinetochores (C) in cells expressing YFP-Astrin (WT, 4A or Δ70) and treated as in (A). Black bars show average across at least two experimental repeats. (D) Representative deconvolved images show Astrin localisation and KT-MT attachment status in monopolar spindles of cells depleted of endogenous Astrin co-expressing Astrin-GFP (WT, 4A or Δ70) and mCherry-GBP-PP1γ. STLC treated cells were exposed to MG132 and either ZM447439 or DMSO prior to immunostaining with antibodies against GFP and mCherry along with CREST anti-sera and stained with DAPI for DNA. Cropped images are areas boxed in white. Absence and presence of Astrin-crescents is highlighted. (E and F) Graphs of percentage of kinetochores with Astrin-GFP crescents (E) or end-on attachments (F) in cells expressing YFP-Astrin (WT, 4A or Δ70) and treated as in (D). Black bars show average across at least two experimental repeats. Percentage of kinetochores with Astrin-GFP sleeves are in *Figure 7—figure supplement 1B* and percentage of lateral and detached kinetochores are in *Figure 7—figure supplement 1C*. '' and 'ns' indicate statistically significant and insignificant differences. Scale as indicated.

The online version of this article includes the following figure supplement(s) for figure 7:

**Figure supplement 1.** Aurora-B inhibition does not rescue the kinetochore enrichment defect of Astrin:PP1-docking mutants.

pool of Astrin-PP1 in this assay. Importantly, time-lapse movies showed that greater than 60% of Astrin depleted cells coexpressing Astrin-WT-GFP and mCherry-GBP-PP1γ displayed unaligned chromosomes (*Figure 8A*; n = 56 cells) and ~54% of anaphase cells displayed lagging chromosomes and a delayed anaphase onset (*Figure 8—figure supplement 1B,C,D*; n = 24 cells). These observations highlight the need for a dynamic interaction between Astrin and PP1. We next analysed the rate of chromosome congression during monopolar to bipolar spindle conversion (a measure of error correction). For this analysis we excluded spindles that were partly bipolar at the beginning of the time-lapse movie (23.7% and 35.7% of cells expressing Astrin alone or both Astrin-GFP and mCherry-GBP-PP1γ, respectively, displayed separated spindle poles). Analysing the rate of chromosome congression in cells presenting clear monopolar spindles showed that Astrin-GFP and mCherry-GBP-PP1γ co-expressing cells were significantly delayed in congressing chromosomes compared to Astrin-GFP alone expressing cells (*Figure 8C*). These findings show that the constitutive delivery of PP1 by Astrin disrupts the dynamic regulation of attachment stability leading to a delay in chromosome congression and anaphase onset and also errors in chromosome segregation.

In summary, two important sequential steps drive the rapid and selective stabilisation of end-on attachments: first, the reduction of Aurora-B activity allows the formation of end-on attachments and the initial recruitment of Astrin-sleeves at the kinetochore. Next, Astrin-mediated delivery of PP1, near the C-terminus of Ndc80, promotes its own enrichment as Astrin-crescents - this positive feedback stabilises end-on attachments to withstand microtubule-mediated pulling (*Figure 9*). Although the inhibition of Aurora-B is essential for the formation of end-on attachment, it is insufficient to stabilise attachments in the absence of Astrin-mediated PP1 delivery. Conversely, constitutive delivery of PP1 by Astrin disrupts the dynamic regulation of KT-MT attachment stability causing delayed chromosome congression and defective anaphase. Thus, Astrin-PP1 interaction acts as a dynamic lock that rapidly and selectively stabilises mature kinetochore-microtubule attachments to ensure the accurate segregation of chromosomes.

## Discussion

Rapid stabilisation of end-on attachments, as soon as they form, is important for maintaining chromosome-microtubule attachments that withstand microtubule-end mediated pulling forces. How cells recognise and selectively stabilise end-on attachments before biorientation is not known. We report the first evidence for a phosphatase-based mechanism that rapidly stabilises kinetochore attachments to microtubule-ends, independent of biorientation. Because Astrin:PP1 dynamically stabilises attachment, independent of biorientation, we propose it as a dynamic 'lock' that works differently from the classical Aurora-B mediated destabilisation of attachments which responds to biorientation status.

Astrin binds selectively to kinetochores attached to microtubule ends (*Shrestha and Draviam, 2013*; *Shrestha et al., 2017*) allowing a selective delivery of Astrin:PP1 onto microtubule-end tethered kinetochores. Without Astrin:PP1, we show that kinetochores can not properly withstand microtubule-pulling or stabilise end-on attachments, leading to an anaphase delay. Importantly, Astrin-mediated PP1 delivery has to occur near the C-terminus of Ndc80 to promote Astrin's own enrichment, presenting evidence for a spatially-restricted positive feedback loop (*Figure 9*). Even in cells

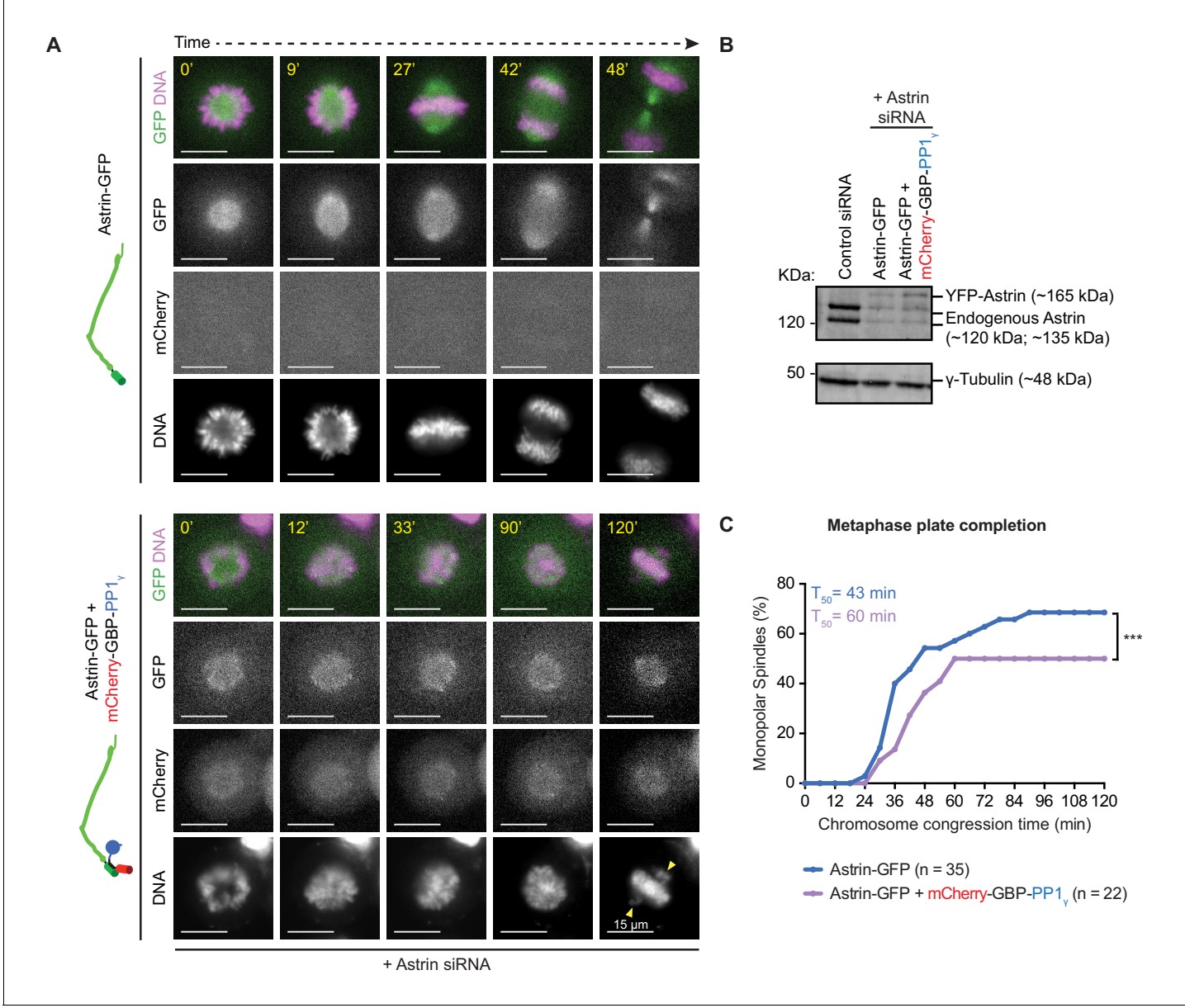

**Figure 8.** Premature constitutive recruitment of Astrin:PP1 to kinetochores disrupts chromosome congression and segregation. (**A**) Time-lapse images show monopolar to bipolar conversion of spindles in Astrin-siRNA treated cells expressing either Astrin-GFP alone or Astrin-GFP and mCherry-GBP-PP1$_\gamma$ as indicated. Yellow arrows mark chromosomes not aligned or congressed during a prolonged metaphase arrest. Scale as indicated. Monopolar to bipolar spindle conversion assay regime outlined in *Figure 8—figure supplement 1A*. (**B**) Immunoblot shows the extent of Astrin depletion following Control or Astrin siRNA treatment as indicated. Lysates were harvested at the end of the time-lapse microscopy shown in A. (**C**) Cumulative graph of percentage of cells with monopolar spindles that aligned or congressed all chromosomes along the metaphase plate, assessed from time-lapse movies of cells as shown in **A**. Only cells that displayed monopolar spindles at the beginning of imaging were considered for analysis. '*' indicates a statistically significant difference in the proportion of monopolar spindles that congress chromosomes within 2 hr following STLC release. Scale as indicated.

The online version of this article includes the following figure supplement(s) for figure 8:

**Figure supplement 1.** Premature and constitutive delivery of Astrin:PP1 disrupts anaphase onset and segregation accuracy.

lacking Aurora-B activity (the attachment destabilising enzyme), Astrin:PP1 is essential for attachment stability, indicating its important role in actively stabilising KT-MT attachments. Supporting this idea, premature constitutive delivery of PP1 via Astrin leads to defective chromosome congression and segregation. Thus, this study reveals an important molecular feature of how end-on attachments

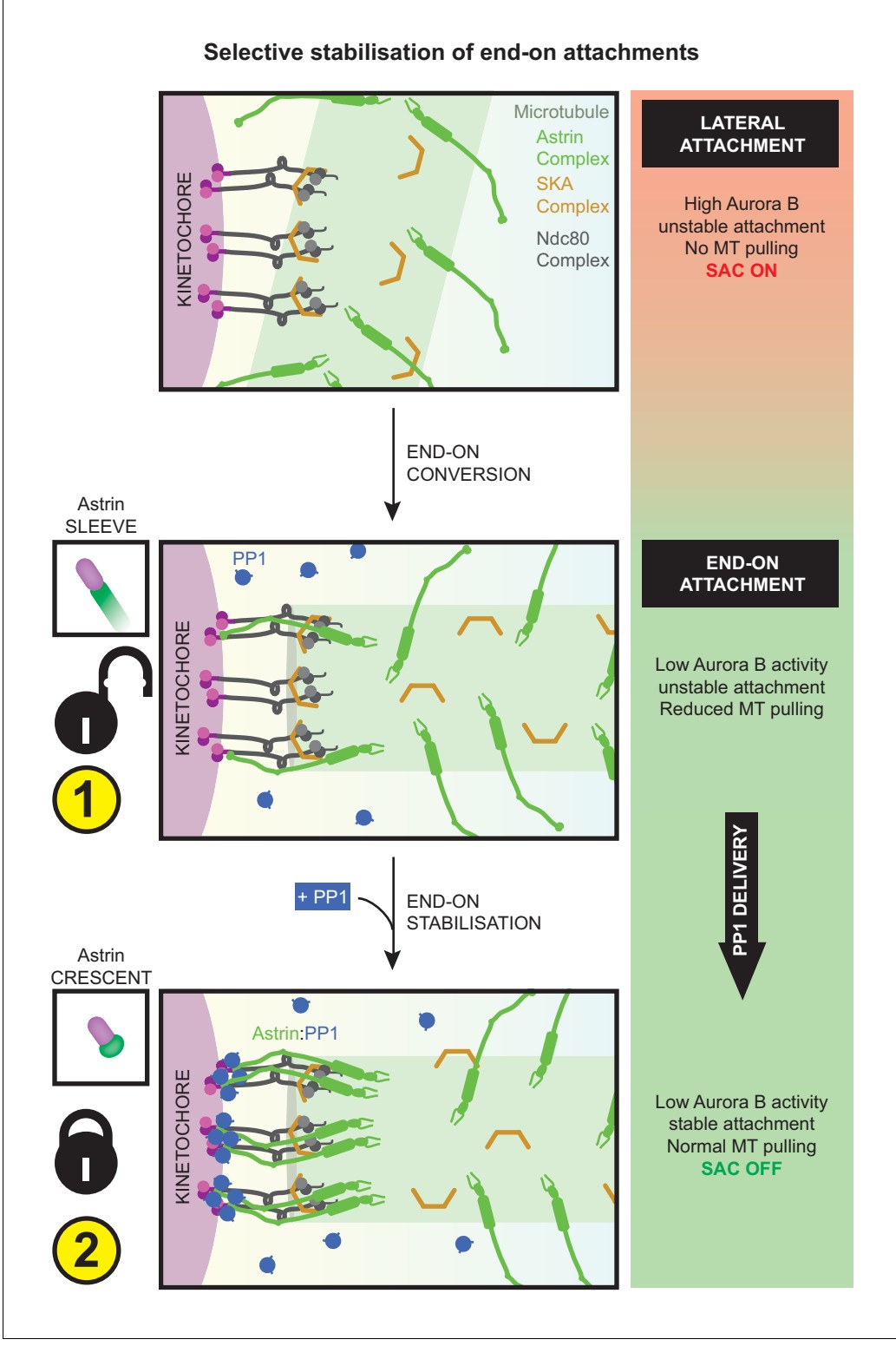

**Figure 9.** Astrin mediates rapid and selective stabilisation of end-on attachments, via local and timely delivery of PP1. Kinetochores are first captured along microtubule-walls (immature lateral attachments) and then attached to microtubule-ends (mature end-on attachments). This process of end-on conversion requires the lowering of Aurora-B at the outer-kinetochore. We show that in the absence of Astrin's tail mediated PP1 interaction, Aurora-B inhibition alone is however insufficient to stably enrich Astrin or maintain end-on attachments. Astrin:PP1 interaction is required to promote Astrin's own enrichment; this positive feedback loop allows rapid stabilisation of

*Figure 9 continued*

end-on attachments. Without Astrin:PP1 interaction, KT-MT attachments become unstable to cold treatment and can not withstand microtubule-mediated pulling forces, triggering the spindle checkpoint and delaying anaphase onset. Conversely, premature constitutive delivery of Astrin:PP1 interferes with error correction, chromosome congression and segregation fidelity. Thus, by dynamically regulating its own enrichment at kinetochores, Astrin-PP1 acts as a spatially restricted 'lock' that recognises and stabilises microtubule interactions, specifically, at end-on attached kinetochores.

are selectively recognised and rapidly stabilised, which is crucial for the normal segregation of chromosomes. In addition, the findings provide a general framework for understanding how a dynamic microtubule-end can escort proteins needed to both recognise and stabilise its presence at a specific subcellular site. Similar phospho-regulation of controlled protein enrichment has been reported at interphase microtubule-ends (*van der Vaart et al., 2011*; *Tamura and Draviam, 2012*).

We propose that the Astrin:PP1 'lock' operates proximal to the conserved tetramer junction between Ndc80-Nuf2 and Spc24-25 (*Valverde et al., 2016*), as targeting PP1 to Astrin's C-terminus, but not it's N-terminus, promotes Astrin enrichment. Using bioinformatics, we identified Astrin's C-terminus as the most evolutionarily conserved region of Astrin across Bilateria (well beyond its recognised presence in Mammalia and Hemichordata; *van Hooff et al., 2017*). These observations shed light on why Astrin's C-terminus (343 a.a) is important for its kinetochore localisation, but not spindle localisation (*Dunsch et al., 2011*; *Kern et al., 2017*). In contrast, Astrin's N-terminus interacts with the microtubule-end binding partner SKAP (*Friese et al., 2016*; *Kern et al., 2017*; *Tamura et al., 2015*). Thus, the C- and N- terminus together allow the Astrin-SKAP complex to bridge the kinetochore and microtubule-end (*Figure 9*). Supporting the significant role of Astrin-Tail in stabilising KT-MT attachments, we could identify Astrin homologs with high conservation in the C-terminal tail elsewhere in Bilateria.

## Both Astrin:PP1 and Aurora-B regulate attachment stability

Aurora-B kinase disallows the lateral to end-on conversion of attachments (*Shrestha et al., 2017*; *Kalantzaki et al., 2015*) and it phosphorylates several outer-kinetochore substrates to reduce their microtubule interaction (*DeLuca et al., 2011*; *Zaytsev et al., 2015*; *DeLuca et al., 2018*; *Wei et al., 2011*; *Long et al., 2017*; *Welburn et al., 2010*). To stabilise end-on attachments, it is essential to reverse Aurora-B-mediated phosphorylation and recruit new outer-kinetochore proteins (*Manning et al., 2010*; *Shrestha et al., 2017*; *Chan et al., 2012*; *Kim et al., 2010*; *Schmidt et al., 2010*; *Cheerambathur et al., 2017*; *Janczyk et al., 2017*; *Kern et al., 2017*; *Umbreit et al., 2014*; *Zhang et al., 2017*). For example, reducing Aurora-B-mediated phosphorylation of Ndc80's N-terminus enhances Ndc80-microtubule binding (*Etemad et al., 2015*; *Tauchman et al., 2015*; *Zaytsev et al., 2015*), promotes Ska recruitment (*Chan et al., 2012*; *Cheerambathur et al., 2017*) and releases MPS1-Ndc80 interaction (*Hiruma et al., 2015*; *Ji et al., 2015*; *Zhu et al., 2013*). Unlike these events that occur near the N-terminus of Ndc80 that interfaces with the microtubule, Astrin-mediated delivery of PP1 is successful only when PP1 is delivered near the C-terminus of Ndc80. In the context of multiple Ndc80 units and SKA cooperatively interacting with the microtubule lattice (*Alushin et al., 2012*; *Janczyk et al., 2017*; *Zaytsev et al., 2013*), Astrin-PP1 recruitment near the C-terminus of Ndc80 can further enhance KT-MT bridges through a different domain of Ndc80. Our study of DSN1 pSer100 status indicates that Astrin-PP1 can modulate phosphorylation at the kinetochore. However, DSN1 phosphorylation per se is not thought to be important for viability (*Welburn et al., 2010*). We speculate that similar phospho-changes proximal to the C-terminus of Ndc80 could be essential to promote Astrin's own enrichment which strengthens KT-MT bridging and rapidly stabilises end-on attachments (*Figure 9*). Together, the Aurora-B pathway and Astrin-PP1 pathway modulate multiple events to destabilise and stabilise end-on attachments, respectively.

Several outer kinetochore pools of PP1 exist. The Astrin:PP1 pool, we report here, mediates a non-overlapping role with those previously reported (details below). PP1 recruited via KNL1 (*Meadows et al., 2011*; *Liu et al., 2010*; *Rosenberg et al., 2011*) is negatively regulated by Aurora-B kinase (*Nijenhuis et al., 2014*); KNL1-bound PP1 is however not critical for stabilising end-on attachments (*Shrestha et al., 2017*). Similarly, PP1 recruited via SKA (*Sivakumar et al., 2016*), a MAP negatively regulated by Aurora-B (*Chan et al., 2012*), is important for spindle checkpoint

silencing but not microtubule-mediated force generation (*Sivakumar et al., 2016*). Cenp-E can also deliver PP1 to the kinetochore and abrogating this delivery blocks chromosome alignment; this again, counteracts Aurora-B-mediated phosphorylation (*Kim et al., 2010*). Unlike these documented roles of PP1 at the outer-kinetochore, Astrin-PP1 mediated attachment stabilisation can not be substituted by Aurora-B inhibition, revealing an important role for Astrin in dictating kinetochore-microtubule attachment stabilisation.

Astrin localises along the spindle, at spindle poles and is enriched at kinetochores (*Mack and Compton, 2001* and this study) with PP1 colocalising predominantly at kinetochores and to a small extent on microtubules. Although the Astrin mutants we report are impaired in their kinetochore but not spindle localisation, and PP1 tethering works only at the C but not the N-terminus of Astrin, we can not rule out that the effect of Astrin C-terminal mutants or PP1 tethered to Astrin's C-terminus may arise from Astrin's role in other subcellular spaces, in addition to kinetochores.

We previously reported that Astrin-SKAP complex is recruited specifically to kinetochores attached to microtubule-ends, but not -walls using fixed-cell studies (*Shrestha and Draviam, 2013*; *Shrestha et al., 2017*). This was confirmed by others (*Kuhn and Dumont, 2017*; *Xu et al., 2014*) consistent with Astrin's recruitment in prometaphase (*Fang et al., 2009*), although other fixed-cell studies (*Kern et al., 2017*; *Mack and Compton, 2001*) indicate a need for biorientation or meta-phase alignment to enrich Astrin-SKAP at kinetochores. Here, we firmly demonstrate using live-cell imaging that Astrin can be enriched at kinetochores of monopolar spindles prior to biorientation. Recognising that Astrin can be recruited to kinetochores before biorientation (in monopolar spin-dles) is highly significant as it supports an inter-kinetochore stretching/tension independent mecha-nism to rapidly stabilise and maintain end-on attachments as soon as they form. Thus, we propose Astrin-PP1 mediated selective stabilisation of end-on attachments as a dynamic lock that works along with Aurora-B, during error correction and end-on conversion of attachments, to ensure the accurate segregation of chromosomes.

## Materials and methods

### Cell culture and drug treatments

HeLa cells (ATCC) cultured in Dulbecco's Modified Eagle's Media was supplemented with 10% FCS and antibiotics (Penicillin and Streptomycin). For live-cell imaging studies, cells were seeded onto 4-well cover glass chambered dishes (Lab-tek; 1064716). The HeLa FRT/TO YFP-Astrin WT, 4A or Δ70 cell lines were conditionally expressing siRNA-resistant YFP-Astrin (WT, 4A or Δ70) was created by transfection of HeLa Flp-In cells with pCDNA5-FRT/TO-YFP-Astrin WT, 4A or Δ70 expression plas-mids followed by a brief Hygromycin selection and sorting for YFP positive using FACS. Both HeLa and Hela Flp-In cell lines were tested and confirmed free of Mycoplasma. Induction of exogenous YFP-Astrin was performed by exposing the cells to DMEM medium supplemented with Teracycline for 48 hr. For inhibition studies, cells were treated with 10 μM Monastrol (1305, TOCRIS), 20 μM STLC (83265,TOCRIS), 10 μM ZM447439 (2458, TOCRIS), 100 nM Taxol (T7191, SIGMA-ALDRICH) or 10 μM MG132 (1748, TOCRIS). For monopolar spindle studies, STLC or Monastrol treatment was for 2 hr and then supplemented with MG132 and ZM447439 for 15' prior to fixation. For bipolar spindle studies, MG132 treatment was for 1 hr (*Hart et al., 2019*).

### Plasmid and siRNA Transfection

siRNA transfection was performed using Oligofectamine according to manufacturer's instructions. siRNA oligos to target Astrin mRNA 5' UTR (GACUUGGUCUGAGACGUGAtt) or Astrin 52 oligo (UCCCGACAACUCACAGAGAAAUU). The former oligo was used in *Figures 5*, *6* and *7* and *Fig-ure 6—figure supplement 1A,B* and the latter oligo was used in the rest of the figures. Negative control siRNA (12,935–300) was from Invitrogen. Plasmid transfection was performed using Turbo-Fect (Fisher; R0531) or DharmaFECT duo (Dharmacom; T-2010) according to manufacturer's instruc-tions. In addition to the standard protocol, after 4 hr of incubation, the transfection medium was removed and fresh selected pre-warmed medium was added to each well. In co-transfection studies, eukaryotic expression vectors encoding Astrin and mCherry-GBP were used in 3:1 ratio. mCherry-GBP-PP1γ expression plasmid was generated by subcloning 7–300 of PP1γ into an mCherry-GBP expression plasmid. Astrin-4A mutant (RVMF to AAAA substitution) was generated using site-

directed mutagenesis. Astrain Δ70 mutant was generated by PCR amplification of 1–1122 coding region of Astrin and subcloning into the desired expression vector. Astrin-GFP (C-terminal fusion) was generated with flexible a.a linker GGGSGGGS between the C-terminus of Astrin and N-terminus of GFP. Plasmid sequences were confirmed by DNA sequencing. Plasmids and plasmid maps are deposited in Ximbio.com.

## Immunofluorescence studies

Cells were cultured on ø13 mm round coverslips (VWR; 631–0150). Unless specified, cells were fixed with ice-cold methanol for a minute. For cold stable assays, 4% PFA was used as in the Super-Resolution Microscopy fixation protocol (*Shrestha et al., 2017*). Following fixation, two quick washes with wash buffer (1X PBS + 0.1% Tween 20) were performed, followed by three washes of 5 min each. Coverslips were incubated with (1X PBS + 0.1% Tween 20 + 1% BSA) for 20 min, before staining with primary antibodies overnight at 4°C. For assessing phospho-DSN1, cells were treated with pre-warmed buffers. Following a quick rinse with PHEM (60 mM PIPES, 25 mM HEPES, 10 mM EGTA, 4 mM MgSO$_4$, pH 6.9), cells were exposed to fixation buffer (4% PFA in PHEM), and then lysed at 37°C for 5 min in PHEM buffer, 1% Triton X-100 and Protease/Phosphatase Inhibitor Cocktail (Cell Signalling Technology; 5872S) before re-exposing to fixation buffer for 20 min. All subsequent washes were performed thrice, with 5 min incubations, in PHEM-T (PHEM buffer + 0.1% Triton X-100) at room temperature. Fixed cells were washed and then blocked with 10% BSA in PHEM buffer for an hour at room temperature prior to overnight incubation at 4°C with primary antibodies diluted in PHEM buffer with 5% BSA, which was followed by a wash prior to incubation with secondary antibodies diluted in PHEM + 5% BSA for 45 min at room temperature. Finally, coverslips were washed with PHEM-T except prior to mounting onto glass slides when coverslips were quickly rinsed in distilled water.

Cells were stained with antibodies against α-Tubulin (Abcam; ab6160; 1:800 or 1:500), GFP (Roche; 1181446001; 1:800), mCherry (Thermo Scientific; M11217; 1:2000), SKAP (Atlas; HPA042027; 1:1000), Astrin (Novus; NB100-74638; 1:1000), SKA3 (Santa-Cruz; sc-390326; 1:500), GFP (Abcam; ab290; 1:1000), mCherry (Abcam; ab167453; 1:2000), ZW10 (Abcam; ab53676; 1:1000), MAD2 (Covance; PRB-452C; 1:500), DSN1 pSer100 (Cheeseman Lab; 1:1000) and CREST antisera (Europa; FZ90C-CS1058; 1:2000) were used. DAPI (Sigma) was used to co-stain DNA. All antibody dilutions were prepared using the blocking buffer. Images of immunostained cells were acquired using 100X NA 1.4 objective on a DeltaVision Core microscope equipped with CoolSnap HQ Camera (Photometrics). Volume rendering (*SoftWorx*) was performed for 3D analysis of KT-MT attachment status (as in *Shrestha et al., 2017*). Deconvolution of fixed-cell images and 3D volume rendering were performed using SoftWorx.

## Immunoblotting studies

Quantitative immunoblotting was performed on proteins separated on 12% SDS-PAGE gels by transferring them overnight onto Nitrocellulose membrane. Membranes were incubated in primary antibodies against Astrin (Proteintech; 14726–1-AP; 1:3000), γ-Tubulin (Sigma-Aldrich; T6793; 1:800), PP1 (Santa-Cruz; sc-7482; 1:5000) and GST (Santa-Cruz; sc-459; 1: 500) for 2 hr and probed using secondary antibodies labelled with infrared fluorescent dyes, which were imaged using an Odyssey imager.

## Microscopy and image analysis

For all high-resolution live-cell imaging assays, cells were either transfected with plasmid vectors 24 hr prior to imaging or directly transferred to imaging in Leibovitz's L15 medium (Invitrogen; 11415064) supplemented with MG132 (1748, TOCRIS; 10 μM) and incubated for 1 hr at 37°C before imaging. For imaging, HeLa and HeLa YFP-PP1$_γ$ cells for FRET measurements, exposures of 0.3 s for the YFP channel and 0.8 s or 1 s for the CFP and FRET channels and at least 6 *Z*-planes, 0.3 μm apart, were acquired using a 100X NA 1.40 oil immersion objective on an Applied Precision DeltaVision Core microscope equipped with a Cascade2 camera under EM mode. Imaging was performed at 37°C using a full-stage incubation chamber set up to allow normal mitosis progression (*Corrigan et al., 2013*) and microtubule dynamics (*Iorio et al., 2015*). For FRET measurements, fluorescence intensities of CFP, YFP and FRET channels were analysed using FIJI (NCBI;

*Schindelin et al., 2012*). For HeLa cells transiently expressing Astrin-CFP or HeLa YFP-PP1$_\gamma$ cells, kinetochores were identified by Astrin or PP1 crescents, respectively. For HeLa YFP-PP1$_\gamma$ cells transiently expressing Astrin-CFP, kinetochores were identified by PP1 crescents. In all conditions, a 0.66 µm$^2$ area was used for measuring the intensity of each signal. For *Figure 4—figure supplement 2*, images were analysed using SoftWorx (GE Healthcare) using a 3 × 3 pixel circle area. Ratios were calculated using Excel (Microsoft) and graphs plotted using Prism6 (GraphPad). Interkinetochore and intercentromeric distances were measured using distance measurement function in Softworx and values plotted using Prism6 (GraphPad).

## Protein expression and purification

pGAT3 vectors with cDNAs encoding His-GST and His-GST-PP1$_\gamma$ (J. Peränen and M. Hyvönen and Blundell groups) were transformed into *E. coli* BL21 cells and grown at 37°C in Luria broth in the presence of 100 µg/µL Ampicillin (A0166, Sigma) to an OD600 of ~0.6. Protein expression was induced by the addition of 1 mM IPTG (I6758, Sigma) for 14 hr at 22°C. Cells were pelleted at 14 K RPM, 4°C for 20 mins and stored at −20°C.

For immobilization on beads, cell pellets were resuspended in lysis buffer (50 mM Tris-HCl, 150 mM KCl, 1% (v/v) TWEEN-20, pH 7.5 supplemented with protease inhibitor cocktail [11873580001, Sigma]), lysed by sonication and cleared by centrifugation at 14 K RPM, 4°C for 30 mins. The cleared supernatant was incubated for an hour with glutathione HiCap Matrix (30900, QIAGEN) at 4°C with continuous rotation and washed with wash buffer (1x PBS supplemented with protease inhibitor cocktail [11873580001, Sigma]). Immobilized proteins were stored at 4°C. All steps were performed on ice unless specified.

## Pulldown assay

HeLa cells seeded in 10 × 15 cm plates were grown to 80% confluency and transfected with control and Astrin siRNA. After 24 hr, cells were arrested with 20 µM STLC for 24 hr. Cells were scraped at room temperature or on ice, washed with wash buffer (1x PBS supplemented with protease inhibitor cocktail [11873580001, Sigma]) and lysed in 1 mL lysis buffer (1x PBST (1x PBS + 0.1% Tween 20), 5 mM Sodium Orthovanadate, 0.02% Triton X-100 supplemented with protease inhibitor cocktail [11873580001, Sigma]) at 75% AMP (Vibracell VCX130, Sonics) for ~15 s. The lysate was cleared of cell debris by centrifugation at 14 K RPM, 4°C for 15 min. Half of the cleared lysate was incubated with either immobilized His-GST-PP1$_\gamma$ or His-GST for an hour on a spinning wheel at 14 RPM, 4°C. Beads were washed four times with 500 µL wash buffer for 5 mins, spun down at 500 RPM (4°C) and resuspended in wash buffer. The samples were analyzed using SDS-PAGE and immunoblotting.

## Astrin-PP1 constitutive binding assay

HeLa cells seeded onto 4-well cover glass chambered dishes were treated with Astrin siRNA (*Shrestha et al., 2017*), incubated for 24 hr and then transfected with plasmid vectors encoding Astrin-GFP alone or co-transfected with mCherry-GBP-PP1$_\gamma$. 24 hr post-transfection of plasmids, cells were arrested for 4 hr in media containing 20 µM STLC supplemented with 0.25 µM SiR-DNA, which was followed by a transient exposure to 10 µM ZM447439 and 10 µM MG132 for 15 min. Cells were quickly washed thrice with warm DMEM, then washed 5 times with 5 min pause using warm DMEM and released in Lebovitz's L15 media. Immediately after the final wash, cells were taken to the microscope and imaged for 2 hr every 3 min and finally, cell lysates were collected for immunoblotting. Lysates were used to probe Astrin depletion levels using Western blotting. For immunofluorescence studies, the same methodology was followed until the release from STLC induced arrest. Cells were fixed 45 min after release using DMEM and stained using antibodies against GFP (for Astrin-GFP), mCherry (for mCherry-GBP-PP1$_\gamma$) and DAPI stain (for DNA).

## Statistical analysis

For fixed-cell images, image analysis was performed using Softworx and Excel (Windows) software. For Astrin kinetochore blinking, image analysis was performed using Fiji (NCBI). Data were plotted using Prism-6 (GraphPad) software. Statistical analysis was performed using Prism6 (GraphPad) or Excel software using data from independent biological replicates, as before (*Zulkipli et al., 2018*). The following statistical tests were performed: Non-parametric Mann-Whitney test: *Figures 1D,*

*2C*, *3B,G* and *4B,C*; *Figure 1—figure supplement 2D*; *Figure 3—figure supplement 3A* and *Figure 3—figure supplement 4B*; *Figure 4—figure supplement 2B*; *Figure 5—figure supplement 2C*; *Figure 5—figure supplement 2C* and *Figure 6—figure supplement 1C*. Non-parametric Kruskal-Wallis H test combined with Dunn's multiple comparisons test: *Figure 5B,C* and *Figure 7B,C,E,F* and *Figure 7—figure supplement 1B and C*. Non-parametric Kolmogorov-Smirnov test: *Figure 8C*.

## Acknowledgements

We would like to acknowledge funding support from BBSRC (R01003X/1), MRC (MR/K50127X/1), QMUL (SBC8DRA2), Islamic Development Bank, and CRUK (C28598/A9787). We thank IM Cheeseman, T Blundell and L Trinkle-Mulchahy for sharing DSN1 antibodies, GST-PP1γ expression vector and HeLa YFP-PP1γ cell lines, respectively. We acknowledge Dr Rajesh Shenoy and Dr Naoka Tamura for the generation of few of the Astrin expression vectors, Revathy Ramalingam, Sam Court and Dr Petra Ungerer for general laboratory support, Steiglitz group and Dr Ruth Rose for Biochemistry support and Dr Isabel Palacios, Dr Lakxmi Subramanian, Dr Peter Thorpe, Dr Gudjon Olaffson and Draviam lab members for comments on the manuscript and discussions on data analysis and acquisition methods. JMD contributed to Figure 3-figure supplement 1, RWP contributed to Figure 3-figure supplement 2, AI contributed to Figure 3-figure supplement 4, Figure 4 - figure supplement 1B and Figure 4F, PG contributed to Figure 4D, 4E and 4F, Figure 4 – figure supplement 1A and 1B and DC contributed to rest of the figures.

## Additional information

### Funding

| Funder | Grant reference number | Author |
|---|---|---|
| Queen Mary University of London | SBC8DRA2 | Viji M Draviam |
| Biotechnology and Biological Sciences Research Council | R01003X/1 | Viji M Draviam |
| Cancer Research UK | C28598/A9787 | Viji M Draviam |
| Medical Research Council | MR/K50127X/1 | Duccio Conti |
| Islamic Development Bank | | Parveen Gul |

The funders had no role in study design, data collection and interpretation, or the decision to submit the work for publication.

### Author contributions

Duccio Conti, Data curation, Formal analysis, Validation, Investigation, Visualization, Methodology; Parveen Gul, Data curation, Formal analysis, Methodology; Asifa Islam, Data curation, Formal analysis, Validation, Methodology; José M Martín-Durán, Formal analysis, Visualization, Methodology; Richard W Pickersgill, Formal analysis, Visualization, Methodology, Writing - original draft related to Figure 3-figure supplement 2; Viji M Draviam, Conceptualization, Formal analysis, Supervision, Funding acquisition, Investigation, Methodology, Writing - original draft, Writing - review and editing

### Author ORCIDs

Duccio Conti (iD) https://orcid.org/0000-0003-4009-5940
Viji M Draviam (iD) https://orcid.org/0000-0001-8295-3689

### Decision letter and Author response

Decision letter https://doi.org/10.7554/eLife.49325.sa1
Author response https://doi.org/10.7554/eLife.49325.sa2

## Additional files

### Supplementary files

• Source data 1. Includes fasta sequences used for bioinformatic search of Astrin elsewhere in Bilateria.

• Supplementary file 1. Key resource table contains the details of all the key reagents used during the development of this work.

• Transparent reporting form

### Data availability

All data generated or analysed during this study are included in the manuscript and supporting files.

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
