## [Decision Letter]

**Acceptance summary:**

Generating stable attachments between kinetochores and microtubules during mitosis is essential for successful chromosome segregation and for maintaining genomic stability. Understanding how cells achieve this feat is a top priority for the field. While previous studies have demonstrated that the kinetochore-associated protein Astrin localizes to mature kinetochore-microtubule attachments and contributes to attachment stability, how it does so remains unclear. In this study, Conti and colleagues demonstrate that Astrin is loaded onto end-on microtubule-bound kinetochores prior to chromosome bi-orientation to promote attachment stability by enriching Astrin's own kinetochore localization and delivering a population of the phosphatase PP1 to kinetochores. The authors find that Astrin-mediated recruitment of PP1 specifically to a region near the Ndc80 C-terminus is required for these activities, as its delivery to other nearby locations does not have the same stabilizing effects. The described mechanisms extend our view of how proper chromosome segregation is achieved and should be of interest to those in the field of mitotic cell division. Determining the PP1 targets in this particular region of the kinetochore that control kinetochore-microtubule attachment stability will be an important and exciting next step forward.

**Decision letter after peer review:**

Thank you for submitting your article "Kinetochores attached to microtubule-ends are stabilised by Astrin bound PP1 to ensure proper segregation of chromosomes" for consideration by *eLife*. Your article has been reviewed by three peer reviewers, and the evaluation has been overseen by Jennifer DeLuca as the Reviewing Editor and Anna Akhmanova as the Senior Editor. The following individual involved in review of your submission has agreed to reveal their identity: Christopher Campbell (Reviewer #1).

The reviewers have discussed the reviews with one another and I have drafted this decision to help you prepare a revised submission.

Summary:

The kinetochore and spindle-associated protein Astrin has previously been implicated in stabilization of kinetochore-microtubule attachments in mammalian cells, however the mechanism by which it does so is poorly understood. In this new study, you report that perturbation of the C-terminus of Astrin disrupts its accumulation at kinetochores and delays anaphase onset. A series of cell-based experiments leads you to conclude that PP1 associates with the C-terminus of Astrin and delivers PP1 to a region near the C-terminus of Hec1/Ndc80 to facilitate kinetochore-microtubule attachment stabilization in an Aurora B-independent manner. The reviewers are in agreement that this study is a significant contribution to the field and the findings could potentially extend the view of how proper and accurate chromosome segregation is achieved. However, the reviewers also agreed that some of the conclusions were not sufficiently supported by the data and several points require clarification.

Overall, the reviewers are supportive of publishing the work in *eLife* after the following Essential Revisions have been addressed.

Essential revisions:

1) The authors identify a PP1 binding motif in the C-terminus of Astrin and find that disruption of this domain has a strong effect on Astrin's localization to the kinetochore. They also report an interaction between Astrin and PP1 based on pull-down and FRET assays, however, it's unclear if the PP1 motif that they mutate is responsible for the interaction that they observe. Testing if the interaction between Astrin and PP1 in these two assays is disrupted by the 4A mutation is required to make the claim that their mutation is affecting the PP1 binding.

2) The evidence that Astrin interacts with PP1 needs to be strengthened. In Figure 4D, the blot for Astrin does not appear to be specific in the GST-PP1 lane. There are at least four additional bands that are pulled down, which may be non-specifically recognized by the Astrin antibody, or non-specifically pulled-down with the bait. To better demonstrate the physical interaction between Astrin and PP1, exogenously-expressed tagged Astrin (e.g. YFP-Astrin) should be used in this experiment, which will likely provide a more specific band.

3) Fusing PP1 to Astrin rescues some aspects of Astrin function but not others (Figure 5A, Figure 6-8). It also appears that the targeting of PP1 to Astrin actually decreases the number of end-on attachments, which is counter to the authors' model that Astrin-PP1 at kinetochores stabilizes end-on attachments. For example, a comparison of WT Astrin in Figures 5B and 7F shows a decrease in end-on attachments from ~60% to ~40% when the GBP-PP1 is added. In the presence of an Aurora B inhibitor (7C vs. 7F), the number of end-on attachments goes from ~90% to ~70% with the targeted PP1. These experiments are complicated by the possibility that artificial fusion of PP1 to Astrin disrupts its spindle functions. The authors should address this by recruiting PP1 to the C-termini of either Ndc80 or Nuf2 in the presence of the 4A or Δ70 mutants, which should not interfere with the spindle microtubule functions of Astrin, and analyze the localization and functions of Astrin, including anaphase timing, end-on attachment stabilization, and chromosome congression.

4) The qualitative descriptions of Astrin localization ("sleeves" vs. "crescents") are somewhat confusing and perhaps over-interpretative. In addition to the current measurements, the authors should quantify the fluorescence of Astrin at kinetochores, using a kinetochore marker as a counterstain. This will be helpful in understanding the levels of Astrin in various kinetochore mutants. Also, the authors' claim that Astrin transitions from "sleeves" to "crescents" is not substantiated. Since it has been shown that a fragment of Astrin that binds kinetochores but not microtubules can localize to kinetochores with proper timing (Dern et al., 2017) it is unclear whether the "sleeve" localization pattern is important for its kinetochore function or simply reflects its spindle localization. Time-lapse imaging of Astrin in living cells would be required to demonstrate that a "transition" occurs.

5) The main focus of the paper concerns phosphatase activity at the kinetochore, therefore it is surprising that there were no experiments looking at the phosphorylation status of kinetochore proteins upon perturbation of the Astrin-PP1 binding site. Considering the authors' claims of PP1 acting via an Aurora B-independent mechanism, immunofluorescence using a phosphoantibody to measure the phosphorylation state of an Aurora B site on a kinetochore protein near the proposed site of Astrin-mediated PP1 recruitment should be carried out. Given the report of an inverse correlation between Dsn1 phosphorylation and Astrin levels at kinetochores (Schmidt et al., 2010), phospho-Dsn1 would be a good antibody to use for this. In addition, identification of a phosphorylation site at the outer kinetochore that is affected by the removal of Astrin-mediated PP1 activity would greatly strengthen the paper. However, given that this is an open-ended question, identification of such a site is not a requirement for manuscript acceptance.

6) Targeting of the non-functional D71N PP1 mutation appears to have a major effect on the ability of WT Astrin to localize to kinetochores (Figure 6—figure supplement 2B). This dominant-negative effect on the WT control calls into question any conclusions made about how the mutants are affected in this experiment.

7) In Figure 1—figure supplement 1B, the recovery rate should be measured. It is unclear if the rate of recovery is different between the spindle and the kinetochore or if it is just the plateau of the recovery that changes.

8) It is difficult to conclude that the targeting of PP1 to Astrin causes a delay in chromosome congression (Figure 8C) when ~20% of cells congress faster than WT and ~20% are slower. Similarly, mitotic progression (Figure 8—figure supplement 1D) appears to be similar to WT for the large majority of cells. Are these differences significant?

9) Is it possible that the effect of the Astrin mutations on Astrin's localization to the kinetochore-microtubule interface is due to Astrin's role at the centrosome or microtubules? Some discussion of this should be included.

10) It is difficult to conceptualize how stabilizing mono-oriented attachments (as observed in the STLC-treated cells) leads to more efficient biorientation. A more detailed explanation of the model here would help. On a related note, the statement "prematurely stabilise attachments leading to a delay in anaphase onset" is counterintuitive as stabilized attachments would silence the checkpoint and speed up anaphase onset.

11) The data in Figure 8 suggesting a "safety lock" are somewhat over-interpreted, as fusion of PP1 to Astrin could also affect its spindle functions as mentioned above. The results suggest that the fusion of PP1 to Astrin may be introducing some defects unrelated to the normal function of Astrin at kinetochores. This caveat should be discussed.

12) The claim that the kinetochore recruitment of Astrin is independent of Aurora B is overstated. Previous studies have demonstrated that inhibition of Aurora B increases the kinetochore intensity of SKAP in STLC-treated cells, suggesting that Aurora B kinase activity negatively regulates SKAP (likely also Astrin) localization at kinetochores. The results in this study also support the negative role of Aurora B in Astrin localization (Figure 7D and Figure 7—figure supplement 1). The basis on which the authors made the claim is that Aurora B inhibition fails to rescue the defects in kinetochore localization of two Astrin mutants (Figure 7A), in which mutations were made in the C-terminal tail. However, if Aurora B's role on Astrin is just through Astrin's C-terminal tail, Aurora B activity would not matter much once the C-terminal tail is removed or mutated. This point should be considered by the authors.

[Editors' note: further revisions were requested prior to acceptance, as described below.]

Thank you for resubmitting your work entitled "Kinetochores attached to microtubule-ends are stabilised by Astrin bound PP1 to ensure proper chromosome segregation" for further consideration by *eLife*. Your revised article has been evaluated by Anna Akhmanova as the Senior Editor and Jennifer DeLuca as the Reviewing Editor.

The manuscript has been improved but there are some remaining issues that need to be addressed before acceptance, as outlined below:

1) In the original manuscript, the authors reported that the C-terminus of Astrin interacts with PP1 through a PP1 docking motif, and mutation of this motif prevents Astrin localization to kinetochores. The reviewers expressed concern that there was no direct evidence that the PP1 motif described in the study is responsible for the interaction they observe, and requested the authors test the Astrin-PP1 interaction using mutants of Astrin (4A and/or Del70) in the FRET and pull-down assays. The authors were not able to pull-down exogenously-expressed Astrin, but found reduced FRET between YFP-PP1 and Astrin-4A-CFP compared to wild-type Astrin (Figure 4—figure supplement 2B). Although the new FRET data are a useful addition, the evidence for interaction through the proposed domain remains weak. One particular concern is that the motif they claim is a PP1 binding motif doesn't conform to the consensus sequence that they cite. From Bollen et al., 2010, "the consensus sequence K/R K/R V/I x F/W, where x is any residue other than Phe, Ile, Met, Tyr, Asp, or Pro" There is a clear conservation of the Met residue at the +3 position in this motif in Astrin, suggesting that it may not bind to PP1. The reviewers understand that demonstrating direct binding using reconstituted components is outside the scope of the current study. In light of this, presenting clear, convincing pulldown data is very important. While the authors attempted to improve upon the original pulldown experiments, there are still some problems. In Figure 4D and Figure 4—figure supplement 1B and C, the bands in the PP1 pulldown lane do not seem to match the input lane, and since a different molecular weight marker is used here and in new data in Figure 4—figure supplement 1C, it is hard to correlate them. A major concern here is that the authors only show a thin strip of limited MW range for the IP with PP1, and the staining appears smeared out over the entire lane, suggesting the pulldown may not be specific. The GST-PP1 pulldown experiment should be repeated with a control and Astrin knockdown. A larger area of the blot needs to be shown (similar to that in Figure 4—figure supplement 1C would suffice), along with its corresponding Ponceau staining to judge pulldown specificity.

2) In the original manuscript, the authors concluded that Astrin-PP1 and Aurora B function in two independent pathways. In the new manuscript, the authors revise this conclusion and report that Aurora B and Astrin-PP1 function in "two separable steps" that act together to control attachment status. Based on the data presented in the manuscript, this claim is not substantiated and needs to be reconsidered for the following reasons. (1) The results in Figure 7—figure supplement 1A demonstrate that Aurora B inhibition results in an increase in Astrin-GFP WT crescent localization (DMSO vs. Aurora B inhibitor), which suggests that Aurora B activity is important for regulating Astrin kinetochore localization. (2) In Figure 7, the results demonstrate that inhibition of Aurora B does not rescue the frequency of end-on attachments or the localization of Astrin to kinetochores in cells expressing Astrin C-terminal mutants. The authors take this to suggest that Astrin stabilizes end-on attachment in a separate manner from Aurora B signaling. The situation is likely much more complex than that. The way the assay is performed, cells are treated with STLC for 2 hours in the presence of different forms of Astrin, and then treated briefly with high-concentrations of Aurora B inhibitors. Thus, kinetochore-attachments are already affected by Astrin perturbation prior to treatment with ZM (i.e. there will be a different distribution of lateral vs. end-on attachments prior to ZM treatment, and it not clear that Aurora B inhibition would promote end-on attachment in this case). Failure in increasing the crescent localization of Astrin mutants and stabilizing end-on attachments upon Aurora B inhibition might simply mean that an Aurora B regulatory domain in Astrin is within the C-terminal domain or the identified four sites. (3) The authors' claim about independency or two separable steps of Astrin:PP1 and Aurora B is partly based on the results produced from their artificial PP1 targeting system. Although this system nicely finds a way for the Astrin mutants to get back to kinetochores independent of Aurora B activity, it is still possible that the artificial system could be so dominant that it overrides many pathways. (4) Their new Dsn1 phosphorylation data suggest that Astrin is a negative regulator of at least one bona fide Aurora B substrate and directly contradicts their model for Astrin and Aurora B acting independently. Overall, a more appropriate interpretation of the data is that Astrin regulates Aurora B targets and other kinase substrates or processes at the kinetochore, and that inhibition of Aurora B alone is not sufficient to rescue Astrin mutant function. This does not necessarily mean that they are separable or don't regulate each other. The text should be modified to consider these points.

3) The original reviews raised concern with the FRAP data presented in Figure 1—figure supplement 2B, and the authors were requested to calculate the recovery rates in the experiment. In response, the authors stated that they could not calculate the rates because they observed no recovery. This is somewhat concerning, since Astrin dynamically loads to kinetochores during mitosis. Such rates have been measured for a number of outer kinetochore proteins, and while they vary, they are indeed measurable. Reporting only plateau levels is not very informative, as they likely reflect the ratio of bound vs. unbound protein instead of the turnover of the bound protein. Given that these data do not contribute significantly to the manuscript, they should be removed. Alternatively, if the authors are able to optimize their assay so that recovery rates can be measured and compared to other outer kinetochore components with published recovery rates, that would certainly be acceptable.

4) In the revised manuscript, there is still little evidence that crescents increase over time. The authors have now actually provided examples where they go back and forth between sleeves and crescents in the new Figure —figure supplement 1. The authors should remove any reference to an increase of crescents over time.

5) The huge decrease in wild-type Astrin KT localization with addition of the F286A;D71N PP1-targeting construct suggests that it is no longer comparable to the functional PP1. The levels of localization are far too different in these two cases. The experiments with the mutant PP1 should either be removed or the authors should state that the dominant-negative effect of the mutations prevents them from drawing any conclusions.

6) Regarding the new Dsn1 phosphorylation data in Figure 5—figure supplement 2. The authors are encouraged to make a scatter plot of the integrated fluorescence intensity of each kinetochore. Currently, the bin sizes are all different making it hard to judge the true effects of the Astrin mutants.

---

## [Author Response]

Essential revisions:1) The authors identify a PP1 binding motif in the C-terminus of Astrin and find that disruption of this domain has a strong effect on Astrin's localization to the kinetochore. They also report an interaction between Astrin and PP1 based on pull-down and FRET assays, however, it's unclear if the PP1 motif that they mutate is responsible for the interaction that they observe. Testing if the interaction between Astrin and PP1 in these two assays is disrupted by the 4A mutation is required to make the claim that their mutation is affecting the PP1 binding.

To test whether Astrin-PP1 interaction is disrupted in the 4A mutant, we compared the extent of FRET between YFP-PP1 and Astrin-CFP WT *versus* Astrin-CFP 4A mutant (outcome of 4 experimental repeats presented in Figure 4—figure supplement 2). We could not perform FRET measurements at all kinetochores (KTs) as the Astrin-4A mutant does not display a sustained kinetochore localisation. Using only the timepoints when the Astrin-4A mutant is brightly localised at the kinetochore, we conclude reduced FRET between YFP-PP1 and Astrin-CFP 4A compared to YFP-PP1γ and Astrin-CFP WT (Figure 4—figure__ supplement 2B).

We attempted several pull-down assays using lysates from HeLa FRT/TO cells conditionally expressing YFP-Astrin -WT or -4A mutant and also lysates from HeLa cells transiently transfected with YFP-Astrin expression plasmids. Unfortunately, we have not been fully successful as exogenously expressed YFP-Astrin is not recovered well in the soluble fraction using our buffer conditions for Astrin-PP1 pull down.

2). The evidence that Astrin interacts with PP1 needs to be strengthened. In Figure 4D, the blot for Astrin does not appear to be specific in the GST-PP1 lane. There are at least four additional bands that are pulled down, which may be non-specifically recognized by the Astrin antibody, or non-specifically pulled-down with the bait. To better demonstrate the physical interaction between Astrin and PP1, exogenously-expressed tagged Astrin (e.g. YFP-Astrin) should be used in this experiment, which will likely provide a more specific band.

To clarify the four bands relating to Astrin in the pull-down lane (Figure 4D), we include a new study of Astrin siRNA treated mitotic cell lysates to identify mitotic forms of Astrin (Figure 4—figure supplement 1C). Two of the four bands in Figure 4D (~120 kDa and 135 kDa) can be seen in the lane containing unsynchronised/interphase cell lysates (Figure 4—figure supplement 1C). Of the other two bands, the one above 135 kDa is specifically found in mitotic cell lysates and lost following Astrin siRNA treatment (Figure 4—figure supplement 1C) confirming it to be a modified mitotic form of Astrin, whereas, the band below 120 kDa (see Figure 4—figure supplement 1B) is enriched in the GST-PP1 pull-down lane and is present in the flowthrough/unbound lysate lanes (i.e., GST-PP1 exposed fractions). Similar four band pattern of Astrin has been reported in Thedieck et al., Cell 2013 (Figure 7G). Lastly, the new Figure 4—figure supplement 1C (immunoblot of control *versus* Astrin siRNA treated cell lysates) shows that our anti-Astrin antibody does not recognise any nonspecific protein in mitotic or interphase whole cell lysates within the 200-95 kDa region. These lines of evidence indicate that all four bands in the pull-down lane of Figure 4D refer to Astrin.

3) Fusing PP1 to Astrin rescues some aspects of Astrin function but not others (Figure 5A, Figure 6-8). It also appears that the targeting of PP1 to Astrin actually decreases the number of end-on attachments, which is counter to the authors' model that Astrin-PP1 at kinetochores stabilizes end-on attachments. For example, a comparison of WT Astrin in Figures 5B and 7F shows a decrease in end-on attachments from ~60% to ~40% when the GBP-PP1 is added. In the presence of an Aurora B inhibitor (7C vs. 7F), the number of end-on attachments goes from ~90% to ~70% with the targeted PP1. These experiments are complicated by the possibility that artificial fusion of PP1 to Astrin disrupts its spindle functions. The authors should address this by recruiting PP1 to the C-termini of either Ndc80 or Nuf2 in the presence of the 4A or Δ70 mutants, which should not interfere with the spindle microtubule functions of Astrin, and analyze the localization and functions of Astrin, including anaphase timing, end-on attachment stabilization, and chromosome congression.

None of the figures indicated above counter our model. Two distinct points need clarification:

1) As previously described in the subsection “KT-MT attachment stability depends on spatially defined delivery of PP1”, compared to Astrin-GFP (C-terminal tag), GFP-Astrin (N-terminal tag) is recruited more robustly (we add Figure 6—figure supplement 1C to further highlight this). Therefore, one cannot compare Figure 5B or Figure 7C against Figure 7F as they differ in the orientation of their tags. The correct controls to ask whether Aurora-B and Astrin-PP1 support distinct steps would be DMSO vs. Aurora-B inhibited samples within the same Figure 7F; statistical significance now added for clarity.

2) Constitutive tethering of PP1 to Astrin-GFP is not expected to rescue all aspects of Astrin function because it disrupts the dynamic nature of interaction between Astrin and PP1. Support for the dynamic interaction between Astrin and PP1 at kinetochore is evident from (i) FRET studies – some but not all KTs display FRET signal (Figure 4A, blue and yellow arrow head) and (ii) delayed mitotic progression and chromosome congression following constitutive Astrin-PP1 interaction (Figure 8 and Figure 8—figure supplement 1). Thus, PP1-tethering studies have been used to get clues about how the C-terminal tail of Astrin works.

We agree that we cannot entirely exclude the impact of Astrin:PP1 on Astrin’s spindle functions. Yet this is less likely as tethering PP1 at Astrin’s N-terminus has no effect, and the effect is seen only when tethered to C-terminus that is positioned close to Ndc80 and away from microtubules (discussed in the subsection “Astrin:PP1 acts as a dynamic ‘lock’ that stabilises attachment and ensures normal anaphase” (also see points 9 and 11 below).Recruiting PP1 or PP1-dead to the C-termini of Ndc80 (Ndc80-GFP) disrupts Astrin localisation at the KT perhaps because of steric hindrance. So, we didn’t pursue this study.

4) The qualitative descriptions of Astrin localization ("sleeves" vs. "crescents") are somewhat confusing and perhaps over-interpretative. In addition to the current measurements, the authors should quantify the fluorescence of Astrin at kinetochores, using a kinetochore marker as a counterstain. This will be helpful in understanding the levels of Astrin in various kinetochore mutants. Also, the authors' claim that Astrin transitions from "sleeves" to "crescents" is not substantiated. Since it has been shown that a fragment of Astrin that binds kinetochores but not microtubules can localize to kinetochores with proper timing (Dern et al., 2017) it is unclear whether the "sleeve" localization pattern is important for its kinetochore function or simply reflects its spindle localization. Time-lapse imaging of Astrin in living cells would be required to demonstrate that a "transition" occurs.

As suggested, to clarify sleeves vs. crescents, we add new figures and expanded legends: i) Figure 1—figure supplement 1 shows transition between sleeves and crescents in multiple KTs, along with change in Astrin intensities at KT relative to a stable centromere marker.

ii)Figure 1A: timelapse images (grey on white) highlight transition from a sleeve (low Astrin intensity at KT that extends to MTs) to a crescent (high Astrin intensity restricted at KT).

iii) Figure 2—figure supplement 1A (Astrin-WT vs. ∆70 mutant intensities at KT with counterstain).

iv) Figure 3—figure supplement 2A (Astrin-WT vs. 4A mutant intensities at KT with counterstain).

v) Expanded Figure 1A legend to explain ‘sleeve’ vs. ‘crescent’ shape and intensity changes.

While sleeve to crescent transitions are clear in our live-cell videos, whether the transition *per se* is essential is not tested here, but it cannot be ruled out. Astrin mutants that bind to KTs but not MTs (highlighted by the reviewer above) show reduced levels of Astrin at metaphase kinetochores (see Figure 3D Kern et al., 2017), suggesting an unrecognised role for Astrin’s microtubule-binding domain in ensuring normal enrichment of Astrin at KTs.

5) The main focus of the paper concerns phosphatase activity at the kinetochore, therefore it is surprising that there were no experiments looking at the phosphorylation status of kinetochore proteins upon perturbation of the Astrin-PP1 binding site. Considering the authors' claims of PP1 acting via an Aurora B-independent mechanism, immunofluorescence using a phosphoantibody to measure the phosphorylation state of an Aurora B site on a kinetochore protein near the proposed site of Astrin-mediated PP1 recruitment should be carried out. Given the report of an inverse correlation between Dsn1 phosphorylation and Astrin levels at kinetochores (Schmidt et al., 2010), phospho-Dsn1 would be a good antibody to use for this. In addition, identification of a phosphorylation site at the outer kinetochore that is affected by the removal of Astrin-mediated PP1 activity would greatly strengthen the paper. However, given that this is an open-ended question, identification of such a site is not a requirement for manuscript acceptance.

As suggested, we tested the levels of Dsn1 phosphorylation in Astrin mutant expressing cells and report that the dephosphorylation of Dsn1 at Ser100 (an evolutionarily conserved Aurora-B phosphorylation site) is compromised (new Figure 5—figure supplement 2).

6) Targeting of the non-functional D71N PP1 mutation appears to have a major effect on the ability of WT Astrin to localize to kinetochores (Figure 6—figure supplement 2B). This dominant-negative effect on the WT control calls into question any conclusions made about how the mutants are affected in this experiment.

In the subsection “KT-MT attachment stability depends on spatially defined delivery of PP1” we clarify that the co-expression of GBP-PP1γ^F286A;D71N^ reduces the proportion of kinetochores enriched with Astrin WT-GFP (Figure 6—figure supplement 2A, B), but it does not fully abolish Astrin-WT localisation at kinetochores, allowing us to investigate relative changes in Astrin mutant localisation. To allow such a study of relative changes, the scoring was binned into cells displaying more or less Astrin crescents. While the coexpression of wildtype-PP1 brings up the proportion of Astrin-crescents in Astrin mutant expressing cells to fully match those observed in Astrin WT expressing cells (i.e., 100% of Astrin positive KTs in both mutant and WT conditions; Figure 6B), the coexpression of non-functional mutant-PP1 does not bring Astrin mutant localisation to match those observed in Astrin WT expressing cells (less than 10% of KTs in mutants *vs.* 50% of KTs in WT expressing cells are Astrin positive; Figure 6—figure supplement 2B). We agree that the dominant negative effect poses a problem, but the relative changes allow us to partially overcome this problem and to indicate the significance of PP1’s activity.

7) In Figure 1—figure supplement 1B, the recovery rate should be measured. It is unclear if the rate of recovery is different between the spindle and the kinetochore or if it is just the plateau of the recovery that changes.

We thank the reviewer for highlighting this point. We reword that the plateau of recovery is different; YFP-Astrin signals at the kinetochore recovered much less compared to YFP-Astrin at the spindle (subsection “Before biorientation, kinetochores bound to MT-ends recruit Astrin”). We are unable to comment on the precise recovery rate because of the very little amount of Astrin recovered (<3%) at the kinetochore.

8) It is difficult to conclude that the targeting of PP1 to Astrin causes a delay in chromosome congression (Figure 8C) when ~20% of cells congress faster than WT and ~20% are slower. Similarly, mitotic progression (Figure 8—figure supplement 1D) appears to be similar to WT for the large majority of cells. Are these differences significant?

Following STLC treatment, some cells retain separated spindle poles. In Figure 8C we had showcased uncongressed to congressed chromosome phenotype independent of spindle organisation phenotype. However, we recognise from the comment above that this causes confusion. So, we now segregate cells with monopolar spindles at the beginning of the video and compare chromosome congression rates in this population (new Figure 8C; analysis explained in the subsection “Astrin:PP1 acts as a dynamic ‘lock’ that stabilises attachment and ensures normal anaphase”). In these cases, constitutive tethering of PP1 to Astrin delays or blocks congression in >60% of cells (extending beyond the 35min average chromosome congression time in Astrin-GFP controls), which is a significant congression delay. In Figure 8—figure supplement 1, mitotic progression is not abrogated but delayed, and more cells exit with lagging chromatids. So, we conclude that tethering PP1 to Astrin can significantly alter mitotic outcomes.

9) Is it possible that the effect of the Astrin mutations on Astrin's localization to the kinetochore-microtubule interface is due to Astrin's role at the centrosome or microtubules? Some discussion of this should be included.

We agree and discuss in the subsection “Astrin:PP1 acts as a dynamic ‘lock’ that stabilises attachment and ensures normal anaphase” (also addressed in points-3 and -11). Although the C-terminal mutants are primarily defective in their kinetochore localisation (Figure 2B and 3A), and the mutants do not disrupt spindle bipolarity (Figure 2D, 2E and Figure 3—figure supplement 2C, D), we cannot exclude that the phenotypes reported include Astrin’s role away from the KT, in other subcellular sites.

10) It is difficult to conceptualize how stabilizing mono-oriented attachments (as observed in the STLC-treated cells) leads to more efficient biorientation. A more detailed explanation of the model here would help. On a related note, the statement "prematurely stabilise attachments leading to a delay in anaphase onset" is counterintuitive as stabilized attachments would silence the checkpoint and speed up anaphase onset.

In the work described in the subsection “Astrin:PP1 acts as a dynamic ‘lock’ that stabilises attachment and ensures normal anaphase”, STLC is washed off before imaging, allowing us to ask whether chromosome congression rates are different in the presence and absence of constitutive PP1 targeted via Astrin. We agree that the sentence in quotes needs clarification and thank the reviewer for spotting it. We clarify that “constitutive delivery of PP1 by Astrin can disrupt dynamic regulation of attachment stability leading to a delay in chromosome congression and anaphase onset and chromosome mis-segregation.”

11) The data in Figure 8 suggesting a "safety lock" are somewhat over-interpreted, as fusion of PP1 to Astrin could also affect its spindle functions as mentioned above. The results suggest that the fusion of PP1 to Astrin may be introducing some defects unrelated to the normal function of Astrin at kinetochores. This caveat should be discussed.

Addressed in points-3 and -9 above (now discussed in the subsection “Astrin:PP1 acts as a dynamic ‘lock’ that stabilises attachment and ensures normal anaphase”). Whether constitutive Astrin:PP1 at KT would impair chromosome congression and segregation efficiency is the question we aimed to address. Based on this study, it’s safe to conclude that Astrin:PP1 interaction needs to be dynamic, but we cannot fully exclude impact from other sites.

12) The claim that the kinetochore recruitment of Astrin is independent of Aurora B is overstated. Previous studies have demonstrated that inhibition of Aurora B increases the kinetochore intensity of SKAP in STLC-treated cells, suggesting that Aurora B kinase activity negatively regulates SKAP (likely also Astrin) localization at kinetochores. The results in this study also support the negative role of Aurora B in Astrin localization (Figure 7D and Figure 7—figure supplement 1). The basis on which the authors made the claim is that Aurora B inhibition fails to rescue the defects in kinetochore localization of two Astrin mutants (Figure 7A), in which mutations were made in the C-terminal tail. However, if Aurora B's role on Astrin is just through Astrin's C-terminal tail, Aurora B activity would not matter much once the C-terminal tail is removed or mutated. This point should be considered by the authors.

Considering this comment and others above, we have reworded the conclusion of Figure 7. Instead of stating Aurora-B and Astrin:PP1 as independent pathways, we indicate them as two separable steps, acting together, to control attachment status (see subsection “Astrin:PP1 and Aurora-B stabilise attachment as two separable steps” for details).

[Editors' note: further revisions were requested prior to acceptance, as described below.]The manuscript has been improved but there are some remaining issues that need to be addressed before acceptance, as outlined below:1) In the original manuscript, the authors reported that the C-terminus of Astrin interacts with PP1 through a PP1 docking motif, and mutation of this motif prevents Astrin localization to kinetochores. The reviewers expressed concern that there was no direct evidence that the PP1 motif described in the study is responsible for the interaction they observe, and requested the authors test the Astrin-PP1 interaction using mutants of Astrin (4A and/or Del70) in the FRET and pull-down assays. The authors were not able to pull-down exogenously-expressed Astrin, but found reduced FRET between YFP-PP1 and Astrin-4A-CFP compared to wild-type Astrin (Figure 4—figure supplement 2B). Although the new FRET data are a useful addition, the evidence for interaction through the proposed domain remains weak. One particular concern is that the motif they claim is a PP1 binding motif doesn't conform to the consensus sequence that they cite. From Bollen et al. 2010, "the consensus sequence K/R K/R V/I x F/W, where x is any residue other than Phe, Ile, Met, Tyr, Asp, or Pro" There is a clear conservation of the Met residue at the +3 position in this motif in Astrin, suggesting that it may not bind to PP1. The reviewers understand that demonstrating direct binding using reconstituted components is outside the scope of the current study. In light of this, presenting clear, convincing pulldown data is very important. While the authors attempted to improve upon the original pulldown experiments, there are still some problems. In Figure 4D and Figure 4—figure supplement 14B and C, the bands in the PP1 pulldown lane do not seem to match the input lane, and since a different molecular weight marker is used here and in new data in Figure 4—figure supplement 1C, it is hard to correlate them. A major concern here is that the authors only show a thin strip of limited MW range for the IP with PP1, and the staining appears smeared out over the entire lane, suggesting the pulldown may not be specific. The GST-PP1 pulldown experiment should be repeated with a control and Astrin knockdown. A larger area of the blot needs to be shown (similar to that in Figure 4—figure supplement 1C would suffice), along with its corresponding Ponceau staining to judge pulldown specificity.

We address two comments here: (a) on citation and (b) Figure 4D.

a) On citation: We had cited Bollen et al., 2010 for the RVxF motif and Wakula et al., 2003 for showing the role of Met in position +3. In Bollen et al., 2010 review, the sentence quoted above refers to work in Hendrickx et al., 2009 which *bioinformatically* analysed all of the then- known RVxF bearing PIPs (PP1-interacting proteins) and did not find Phe, Ile, Met, Tyr, Asp, or Pro in position +3. However, mutagenesis of the RVTF motif in NIPP1 shows that “the penultimate position of the RVxF motif.… can be held by any residue except Pro” (Wakula et al., 2003). In agreement, our model of Astrin’s PP1 docking peptide using the structure of a KNL1 peptide bound to PP1 (Bajaj et al., 2018) shows the +3 position of the RVxF motif is exposed and can be occupied by any a.a except Proline (Figure 3—figure supplement 2). For the sake of clarity, we omit previous citations and include Bajaj et al., 2018 as its recent and relevant.

b) Clarity of Figure 4 As suggested, we repeated our pull-down with Control and Astrin siRNA treated cells and we show that Astrin can be pull-down specifically with GST-PP1 but not GST, and Astrin-PP1 (Figure 4E). Please note that bands in input and pull-down lane cannot be directly compared as purified PP1γ is active and may dephosphorylate interactors upon prolonged exposure. We include matched molecular weight markers for the pulldown studies. In addition, we a) quantify Astrin intensities in lanes corresponding to GST-PP1 or GST, both before and after exposure to mitotic cell lysates from two repeats (Figure 4F) and b) probe for γ-tubulin as a positive control in Astrin siRNA treated lysates.

2) In the original manuscript, the authors concluded that Astrin-PP1 and Aurora B function in two independent pathways. In the new manuscript, the authors revise this conclusion and report that Aurora B and Astrin-PP1 function in "two separable steps" that act together to control attachment status. Based on the data presented in the manuscript, this claim is not substantiated and needs to be reconsidered for the following reasons. (1) The results in Figure 7—figure supplement 1A demonstrate that Aurora B inhibition results in an increase in Astrin-GFP WT crescent localization (DMSO vs. Aurora B inhibitor), which suggests that Aurora B activity is important for regulating Astrin kinetochore localization. (2) In Figure 7, the results demonstrate that inhibition of Aurora B does not rescue the frequency of end-on attachments or the localization of Astrin to kinetochores in cells expressing Astrin C-terminal mutants. The authors take this to suggest that Astrin stabilizes end-on attachment in a separate manner from Aurora B signaling. The situation is likely much more complex than that. The way the assay is performed, cells are treated with STLC for 2 hours in the presence of different forms of Astrin, and then treated briefly with high-concentrations of Aurora B inhibitors. Thus, kinetochore-attachments are already affected by Astrin perturbation prior to treatment with ZM (i.e. there will be a different distribution of lateral vs. end-on attachments prior to ZM treatment, and it not clear that Aurora B inhibition would promote end-on attachment in this case). Failure in increasing the crescent localization of Astrin mutants and stabilizing end-on attachments upon Aurora B inhibition might simply mean that an Aurora B regulatory domain in Astrin is within the C-terminal domain or the identified four sites. (3) The authors' claim about independency or two separable steps of Astrin:PP1 and Aurora B is partly based on the results produced from their artificial PP1 targeting system. Although this system nicely finds a way for the Astrin mutants to get back to kinetochores independent of Aurora B activity, it is still possible that the artificial system could be so dominant that it overrides many pathways. (4) Their new Dsn1 phosphorylation data suggest that Astrin is a negative regulator of at least one bona fide Aurora B substrate and directly contradicts their model for Astrin and Aurora B acting independently. Overall, a more appropriate interpretation of the data is that Astrin regulates Aurora B targets and other kinase substrates or processes at the kinetochore, and that inhibition of Aurora B alone is not sufficient to rescue Astrin mutant function. This does not necessarily mean that they are separable or don't regulate each other. The text should be modified to consider these points.

As recommended, we remove conclusions on whether Aurora-B and Astrin-PP1 pathways are independent or not. Instead, we simply state that “Astrin-PP1 and Aurora-B control KT-MT attachment status.” Although the phospho-DSN1 signals we study do not influence KT-MT attachments or cell viability, we agree with the reviewer that we do not fully demonstrate how the two pathways act separately to control KT-MT attachment status.

3) The original reviews raised concern with the FRAP data presented in Figure 1—figure supplement 2B, and the authors were requested to calculate the recovery rates in the experiment. In response, the authors stated that they could not calculate the rates because they observed no recovery. This is somewhat concerning, since Astrin dynamically loads to kinetochores during mitosis. Such rates have been measured for a number of outer kinetochore proteins, and while they vary, they are indeed measurable. Reporting only plateau levels is not very informative, as they likely reflect the ratio of bound vs. unbound protein instead of the turnover of the bound protein. Given that these data do not contribute significantly to the manuscript, they should be removed. Alternatively, if the authors are able to optimize their assay so that recovery rates can be measured and compared to other outer kinetochore components with published recovery rates, that would certainly be acceptable.

We have removed this supplementary data.

4) In the revised manuscript, there is still little evidence that crescents increase over time. The authors have now actually provided examples where they go back and forth between sleeves and crescents in the new Figure 1—figure supplement 1. The authors should remove any reference to an increase of crescents over time.

We have removed reference to increase over time.

5) The huge decrease in wild-type Astrin KT localization with addition of the F286A;D71N PP1-targeting construct suggests that it is no longer comparable to the functional PP1. The levels of localization are far too different in these two cases. The experiments with the mutant PP1 should either be removed or the authors should state that the dominant-negative effect of the mutations prevents them from drawing any conclusions.

As suggested, we have removed this data.

6) Regarding the new Dsn1 phosphorylation data in Figure 5—figure supplement 2. The authors are encouraged to make a scatter plot of the integrated fluorescence intensity of each kinetochore. Currently, the bin sizes are all different making it hard to judge the true effects of the Astrin mutants.

Scatter plot now added as Figure 5—figure supplement 2C.